# Microencapsulation and nanowarming enables vitrification cryopreservation of mouse preantral follicles

Conghui Tian[1], Lingxiao Shen[1], Chenjia Gong [2], Yunxia Cao[3,4], Qinghua Shi [2] ✉ & Gang Zhao [1] ✉

Preantral follicles are often used as models for cryopreservation and in vitro culture due to their easy availability. As a promising approach for mammalian fertility preservation, vitrification of preantral follicles requires high concentrations of highly toxic penetrating cryoprotective agents (up to 6 M). Here, we accomplish low-concentration-penetrating cryoprotective agent (1.5 M) vitrification of mouse preantral follicles encapsulated in hydrogel by nanowarming. We find that compared with conventional water bath warming, the viability of preantral follicles is increased by 33%. Moreover, the cavity formation rate of preantral follicles after in vitro culture is comparable to the control group without vitrification. Furthermore, the percentage of MII oocytes developed from the vitrified follicles, and the birth rate of offspring following in vitro fertilization and embryo transfer are also similar to the control group. Our results provide a step towards nontoxic vitrification by utilizing the synergistic cryoprotection effect of microencapsulation and nanowarming.

In recent years, an increasing number of malignant diseases (such as breast cancer and hematological diseases) have occurred in young women[1]. Although active chemo- and radiotherapy or bone marrow transplantation can cure 90% of girls and young women, these treatments still damage the gonadal glands and even lead to premature ovarian failure[2-5]. In addition, due to work or financial reasons, many women who postpone childbearing are affected by infertility[6]. As a result, the demand for fertility preservation in women is dramatically increasing[7]. At present, methods to preserve female fertility mainly include cryopreservation of embryos, unfertilized oocytes[8] and ovarian tissue[9,10]. Among them, embryo cryopreservation requires women to have a male partner or use donated sperms, which may involve

religious, moral, ethical and other issues[11]. Oocyte cryopreservation is mainly used by women who have reached adolescence and can withstand multiple rounds of ovarian stimulation[12]. For prepubescent girls and women in urgent need of cancer treatment, cryopreservation of ovarian tissue containing healthy follicles is the only option[13]. However, this method is not suitable for women with certain malignant diseases (such as acute leukemia) because malignant cells may migrate to the ovarian tissue to be transplanted in the future[14]. Even worse, many follicles may be lost due to delays and failures in vascular reconstruction[15], and qualitative and quantitative assessment of follicles in ovarian tissue is almost impossible[16]. For these women, cryopreservation of isolated ovarian follicles and in vitro culture is an

[1]Department of Electronic Engineering and Information Science, University of Science and Technology of China, Hefei, China. [2]Division of Reproduction and Genetics, The First Affiliated Hospital of USTC, Hefei National Laboratory for Physical Science at Microscale, the CAS Key Laboratory of Innate Immunity and Chronic Disease, School of Basic Medical Sciences, Division of Life Sciences and Medicine, Biomedical Sciences and Health Laboratory of Anhui Province, CAS center for Excellence in Molecular Cell Science, Collaborative Innovation Center of Genetics and Development, University of Science and Technology of China, Hefei, China. [3]Department of Obstetrics and Gynecology, The First Affiliated Hospital of Anhui Medical University, Hefei, China. [4]NHC Key Laboratory of Study on Abnormal Gametes and Reproductive Tract (Anhui Medical University), Anhui Provincial Engineering Research Center of Biopreservation and Artificial Organs, Hefei, China. ✉e-mail: qshi@ustc.edu.cn; zhaog@ustc.edu.cn

alternative and safe option[17]. Preantral follicles (PAFs) are the early stage of follicles, which are composed of an oocyte surrounded by one or few layers of granulosa cells[18]. PAFs were abundant and easily available in hormone-stimulated prepubertal mice or young adolescent female mice[19]. Therefore, mouse PAFs were often used as models for optimizing cryopreservation, in vitro culture conditions, and fertility preservation in fertility studies[18,19].

Vitrification (ice-free) is a method for cryopreservation of PAFs that is considered more effective than traditional slow freezing[20]. However, as described previously high concentrations of highly toxic (at body temperature) penetrating cryoprotective agents (pCPAs; up to 6 M) are usually required to achieve vitrification of PAFs[21,22]. This factor will inevitably lead to multistep washing of the PAFs to remove the toxic pCPAs, potentially placing cells at risk of osmotic shock and causing cell loss[23]. Therefore, low-concentration-penetrating cryoprotective agent (low-pCPA) vitrification cryopreservation of PAFs is urgently needed. Currently, devitrification/recrystallization during the warming process is the main obstacle to low-pCPA vitrification of biological samples, as this process can cause fatal damage to cells[24]. Generally, there are two ways to solve the above problem: one is the active ice inhibition approach which involves introducing physical field-assisted space heating techniques (e.g., electromagnetic heating, laser heating)[25,26], and the other is the passive ice suppression approach mediated by developing various types of ice recrystallization inhibitors (IRIs) (e.g., antifreeze proteins[27], synthetic polymer PVA, aryl glycoside IRI analogs, OQCN quantum dots[28–34]). However, their ice suppressive capacity is not tunable once introduced before freezing. Physical field-assisted heating technology uses responsive nanoparticles (NPs) that can absorb the energy of the external physical field and convert it into heat energy to enable rapid and uniform heating of biological samples[35]. The heating rate can be adjusted flexibly by changing some parameters (e.g., current and voltage), thereby reducing thermal stress damage[36]. Although some NPs have been widely used in cryopreservation, there is still a risk of cell entry after long-term direct contact[37]. To improve the heating efficiency, researchers generally increase the amounts of NPs or the power of the external physical field. However, increasing the concentration of NPs may cause the aggregation of nanomaterials, affecting the heating effect. An increase in the power of the physical field is more difficult to control and can easily cause the sample to overheat[38]. In terms of removal, some NPs are easy to remove (e.g., magnetic $Fe_3O_4$ NPs)[37], while others are more difficult to remove [e.g., photothermal materials such as gold NPs and graphene oxide (GO)].

Studies have shown that the hydrogel microencapsulation strategy of cells provides a protective barrier to cells, reducing extracellular ice damage and preventing ice nuclei from growing in cells[39]. In addition, hydrogel microencapsulation was shown to effectively reduce the demand for pCPAs during vitrification by minimizing ice crystal growth during cooling and warming[24,38]. In addition to cryopreservation, hydrogel microencapsulation provides a three-dimensional (3D) microenvironment similar to the extracellular matrix for PAF culture in vitro[40,41]. 3D culture can maintain the spherical structure of PAFs and the connection between oocytes and granulosa cells, which is essential for the development of PAFs in vitro[42]. Therefore, PAFs encapsulated by hydrogels can be observed as a whole for freezing, thawing and in vitro culture.

In this study, we successfully accomplished low-pCPA (1.5 M) vitrification via hydrogel microencapsulation and nanowarming, using mouse PAFs as a model. Up to 90% of the PAFs survived after cryopreservation. Due to the use of hydrogel microencapsulation, NPs were completely separated from PAFs, thus eliminating the potential toxicity to PAFs. Furthermore, these frozen PAFs could develop into preovulatory antral follicles in vitro (in hydrogel microcapsules previously used for cryopreservation) after 13 days and mature oocytes could be obtained. These mature oocytes developed to the blastocyst stage in vitro after in vitro fertilization (IVF), and zygotes also developed into mouse pups after transferring 2-cell stage embryos to pseudopregnant recipients. Finally, the mouse pups produced a healthy second generation through natural mating after sexual maturity. Thus, our results may provide valuable guidance for cryopreservation of PAFs, facilitating the preservation of female fertility in the clinic.

## Results

### Microencapsulation, cryopreservation, in vitro development of PAFs, IVF and transplant

A schematic diagram of the microencapsulation, cryopreservation and subsequent development of PAF is shown in Fig. 1. PAF-encapsulated hydrogel microcapsules were generated by dropping a 1% (w/v) alginate solution containing PAFs into a 0.15 M $CaCl_2$ solution through a centrifugal microfluidic device (Fig. 1a and Supplementary Movie 1). The particle size of the hydrogel microcapsules varied slightly at different centrifugation rates. Moreover, the morphology of the lyophilized hydrogel microspheres did not change before and after freezing (Supplementary Fig. 1). The PAF-encapsulated hydrogel microcapsules were transferred into CPA (0.75 M EG + 0.75 M PROH + 1 M trehalose) supplemented with NPs (0.3% $Fe_3O_4$ and 0.03% GO) and incubated at 4 °C for 10–15 min. Aspirate them into a 2.5 mL straw and seal the straw with wax oil. The straw was then quickly immersed in liquid nitrogen ($LN_2$) for vitrification. After 24 h, the straw was removed from $LN_2$ and quickly immersed in a 50 mL centrifuge tube filled with 37 °C water and placed in an electromagnetic coil, while the laser transmitter was turned on for rapid warming (Fig. 1b). After thawing, the solution in the straw was transferred to a 6 mm dish containing washing solution (MEM medium supplemented with 10% FBS). $Fe_3O_4$ nanoparticles (NPs) in the solution were drawn to the edge of the dish by a magnet and then removed using a pipette (Fig. 1b). The hydrogel microcapsules were siphoned and transferred to a 96-well plate for another wash. The hydrogel microcapsules were then transferred to a 96-well plate via a siphon and subjected to one more wash. The PAFs were cultured in 96-well plates and developed into antral follicles on day 13. Next, the hydrogel microcapsules were dissolved by 75 mM sodium citrate and antral follicles were collected. Antral follicles were punctured via a syringe needle to release the cumulus-oocyte complexes (COCs). Oocytes and granulosa cells were separated by hyaluronidase (Fig. 1c). MII stage oocytes were selected for IVF. 2-cell stage embryos were transplanted into pseudopregnant recipient and produced healthy offspring (Fig. 1d).

### Thermal effect of $Fe_3O_4$ NPs and GO NPs

Typical pictures of cooling and warming using PSs W or W/O NPs are displayed in Fig. 2a. The characterization of $Fe_3O_4$ and GO NPs is shown in Supplementary Fig. 2. PSs #1–#4 containing CPAs are shown in Fig. 2a (i). During cooling, PSs #1–#4 were translucent, and no visible ice could be found. Thus, they were all vitrified in liquid nitrogen ($LN_2$) [Fig. 2a (ii)]. However, devitrification/recrystallization occurred in PSs during warming [Fig. 2a (v)]. Typical pictures of ice crystals occurring, growing to maximum recrystallization and ice crystals thawing were obtained. The warming of PS #1 in the water bath is demonstrated in Supplementary Movie 2. Compared with those of PS #1, the ice crystal residence time was shorter in PS #2 with MIH and PS #3 with LIH (4.2 s vs. 2.96 s and 2.92 s, Supplementary Movies 2, 3, 4). As expected, PS #4 with MIH and LIH showed the shortest ice crystal duration (2.40 s, Supplementary Movie 5). Therefore, this will shorten the residence time in the harmful temperature range during the warming process, thus reducing ice crystal damage to cells.

Nanowarming combining MIH and LIH may minimize devitrification/recrystallization and improve the quality of PAFs after low-pCPA vitrification and warming. For analysis of the effect of MIH and LIH, the thermal history of CPA solution containing GO and $Fe_3O_4$ NPs during warming in the case of MIH and LIH is depicted in Fig. 2a (iii).

The fastest warming rate (-93 °C/s, calculated from −196 °C to −14 °C) was found when the MIH and LIH were used together, suggesting that the combination of MIH and LIH had the best heat transfer effect [Fig. 2a (**iv**)].

In addition, the propagation and penetration depth of the laser inside DI water, carbon black suspension and NP suspension were used to test the LIH effect of NPs (Fig. 2b). Perhaps because of the absorption and scattering of the laser by solutions, the laser became increasing weaker with increasing solution depth. The effective depth of the laser in the NP suspension was much greater than that in the carbon black suspension, which indicated that the LIH effect of the NP suspension was good. In addition, the change in temperature of the CPA solution in a 1.5 mL centrifugal tube from 36.6 °C under MIH and LIH was recorded by infrared thermography (Fig. 2c). The temperature of the CPA solution with $Fe_3O_4$ and GO NPs (0.3%, 0.03%, w/v) under MIH and LIH rose faster than that using either MIH or LIH alone. Moreover, compared with that of the CPA solution with $Fe_3O_4$ and GO NPs (0.4%, 0.04%, w/v) only under MIH or LIH, the temperature of the

CPA solution with $Fe_3O_4$ and GO NPs (0.3%, 0.03%, w/v) under MIH and LIH was higher. Therefore, the combined application of MIH and LIH has a substantial heating effect. In addition, the biocompatibility of NPs was tested, as shown in Supplementary Fig. 3. The results showed that the survival rates of PAFs with (W/) or without (W/O) encapsulation in the medium with (W/) or without (W/O) NPs were basically similar after incubation for different times (0, 6, 12, 24, 36, 48, 60 h) at 37 °C. This suggested that neither NPs nor hydrogels were toxic to PAFs. In addition, energy dispersive X-ray spectroscopy (EDS) of the hydrogel microspheres was performed to examine whether the NPs could be completely separated from PAFs. The results showed that carbon, oxygen, calcium and iron elements could be observed on the surface of the hydrogel microspheres after incubation with NPs, but no iron element can be detected inside the hydrogel. After washing, the iron element could no longer be observed on the surface and inside of the hydrogel microspheres (Supplementary Fig. 4a). This indicated that some $Fe_3O_4$ NPs adhered to the surface of the hydrogel microspheres, but could not enter the interior of the hydrogel after

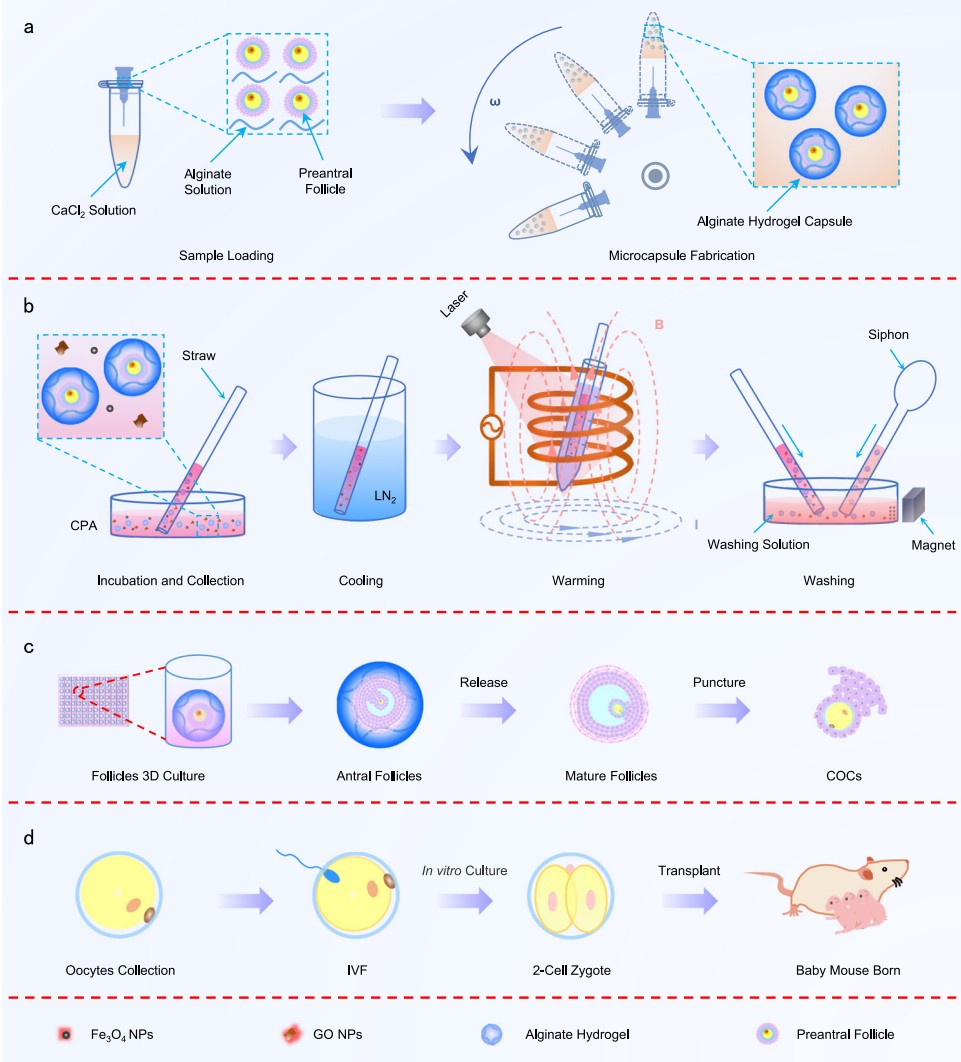

**Fig. 1 | Schematic illustration of microencapsulation, cryopreservation, development of preantral follicles (PAFs) in vitro and subsequent in vitro *fertilization* (IVF) and transplant. a** Fabrication of PAF-alginate hydrogel microcapsules by centrifugal microfluidic technology. $CaCl_2$ solution: 0.15 M $CaCl_2$ solution; Alginate solution: 1% (w/v) alginate solution. **b** Cryopreservation and nanowarming [combined with magnetic induction heating (MIH) and laser-induced heating (LIH)] of PAFs. CPA: 0.75 M ethylene glycol (EG) + 0.75 M 1, 2-propanediol (PROH) + 1 M trehalose; Washing solution: α-MEM + 10% FBS. **c** In vitro culture of PAFs after warming. **d** IVF and the birth of the next generation of mice after in vivo transplantation of 2-cell zygotes. NPs nanoparticles, GO graphene oxide, COCs cumulus-oocyte complex, $LN_2$ liquid nitrogen, CPA cryoprotective agent.

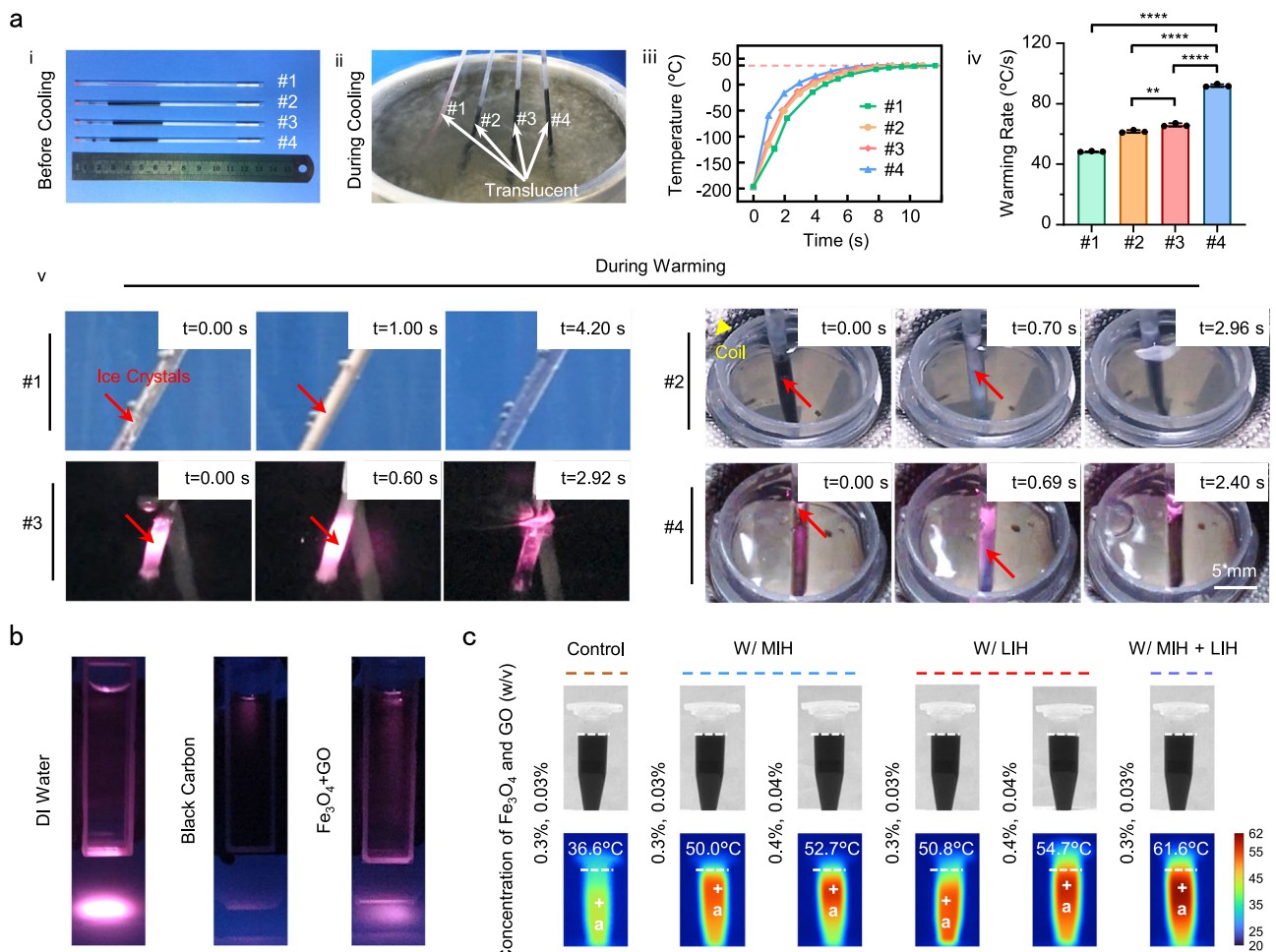

**Fig. 2 | Cooling and warming of CPAs in plastic straws and the thermal effect of NPs excited by near-infrared and magnetic fields. a** Typical images display the cooling and warming of CPAs in plastic straws (PSs) and the thermal history of the nanowarming process. $P_{\#1\text{-}\#4}$ = 2.4980e−12, $P_{\#2\text{-}\#4}$ = 7.0018e−10, $P_{\#3\text{-}\#4}$ = 2.7421e −09, $P_{\#2\text{-}\#3}$ = 0.0013 (iv). **b** Scattered light images of 0.3% (w/v) $Fe_3O_4$ and 0.03% (w/ v) GO NP suspension, 0.1% (w/v) black carbon suspension and deionized (DI) water made by a camera (D5500, Nikon, Japan). **c** Infrared thermogram of CPA solution containing $Fe_3O_4$ and GO NPs under MIH and LIH. #1: CPA (0.75 M EG, 0.75 M PROH,

1 M trehalose), 37 °C water bath; #2: CPA with NPs (0.3% $Fe_3O_4$ and 0.03% GO NPs, w/v), 37 °C water bath and MIH; #3: CPA with NPs (0.3% $Fe_3O_4$ and 0.03% GO NPs, w/ v), 37 °C water bath and LIH; #4: CPA with NPs (0.3% $Fe_3O_4$ and 0.03% GO NPs, w/v), 37 °C water bath and MIH + LIH. One-way analysis of variance (ANOVA) and Tukey's post hoc were used for statistical analysis. ns: $p > 0.05$; **$p \leq 0.01$; ****$p \leq 0.0001$. Control: without heating; W/ MIH: with magnetic induction heating; W/ LIH: with laser-induced heating; W/ MIH + LIH: with magnetic induction heating and laser-induced heating. Data are presented as the mean ± standard deviation (SD) (**a** (iv)).

incubation with NPs. Moreover, the $Fe_3O_4$ NPs could be washed away completely. In addition, the TEM results of PAFs also showed that no $Fe_3O_4$ NPs were took in by PAFs (Supplementary Fig. 4b, c). From all the above results, it could be seen that the PAFs and NPs were separated by the hydrogel microspheres.

### Viability of PAFs after cryopreservation

PAFs will suffer irreversible osmotic damage under high concentrations of CPAs, so the concentration of pCPAs was substantially reduced in this study compared to that of traditional vitrification cryopreservation of PAFs[43]. Since trehalose can increase cell viability after vitrification[44], 1 M trehalose was added to the CPAs. Typical DIC and fluorescence images of PAFs after warming W/ or W/O MIH or LIH are exhibited (Fig. 3a). There were no statistical differences in the survival rate of PAFs among the different CPA concentration groups. However, the value of PAF survival rate with 1.5 M CPA was slightly higher than that with 1 M and 2 M CPA (Fig. 3b). Thus, 1.5 M CPA was an appropriate choice for cryopreservation of PAFs. When PAFs were encapsulated in alginate hydrogel, they had a higher viability than unencapsulated PAFs. Furthermore, PAF survival increased with increasing hydrogel concentration (Fig. 3c). It is possible that freezable water is reduced by

hydrogels and that ice crystal formation is prevented during cooling and warming. Given that the low concentration of alginate hydrogel was more conducive to the development of PAFs in vitro[40], 1% (w/v) alginate hydrogel and 1.5 M CPA were chosen for the following experiments. When $Fe_3O_4$ and GO NPs (0.3% and 0.03%, w/v) were mixed into 1.5 M CPA with a 15 A current intensity, the PAFs encapsulated in alginate hydrogels showed increased survival compared with those W/O MIH (70% vs. 57%). In addition, when 1.5 M CPA with NPs under 3 W/cm² light intensity was used, the viability of PAFs was higher than that W/ MIH (79% vs. 70%) (Fig. 3d). However, the survival rate was still not very high; the single use of MIH or LIH may not have been sufficient to overcome the formation of ice crystals during warming, leading to devitrification and intracellular ice damage. To accelerate the warming rate and minimize the formation of ice crystals during warming, we used MIH and LIH together. When 1.5 M CPA with NPs under a 15 A current intensity and 3 W/cm² light intensity was used, PAF survival increased significantly (90%) (Fig. 3d). Thus, the combination of MIH and LIH significantly improved the warming efficiency and reduced the formation of ice crystals. In addition, PAF survival rates changed with different current intensities (10 A, 15 A, 20 A) and different combinations of $Fe_3O_4$ and GO NPs (0.2% and 0.03%, 0.3% and

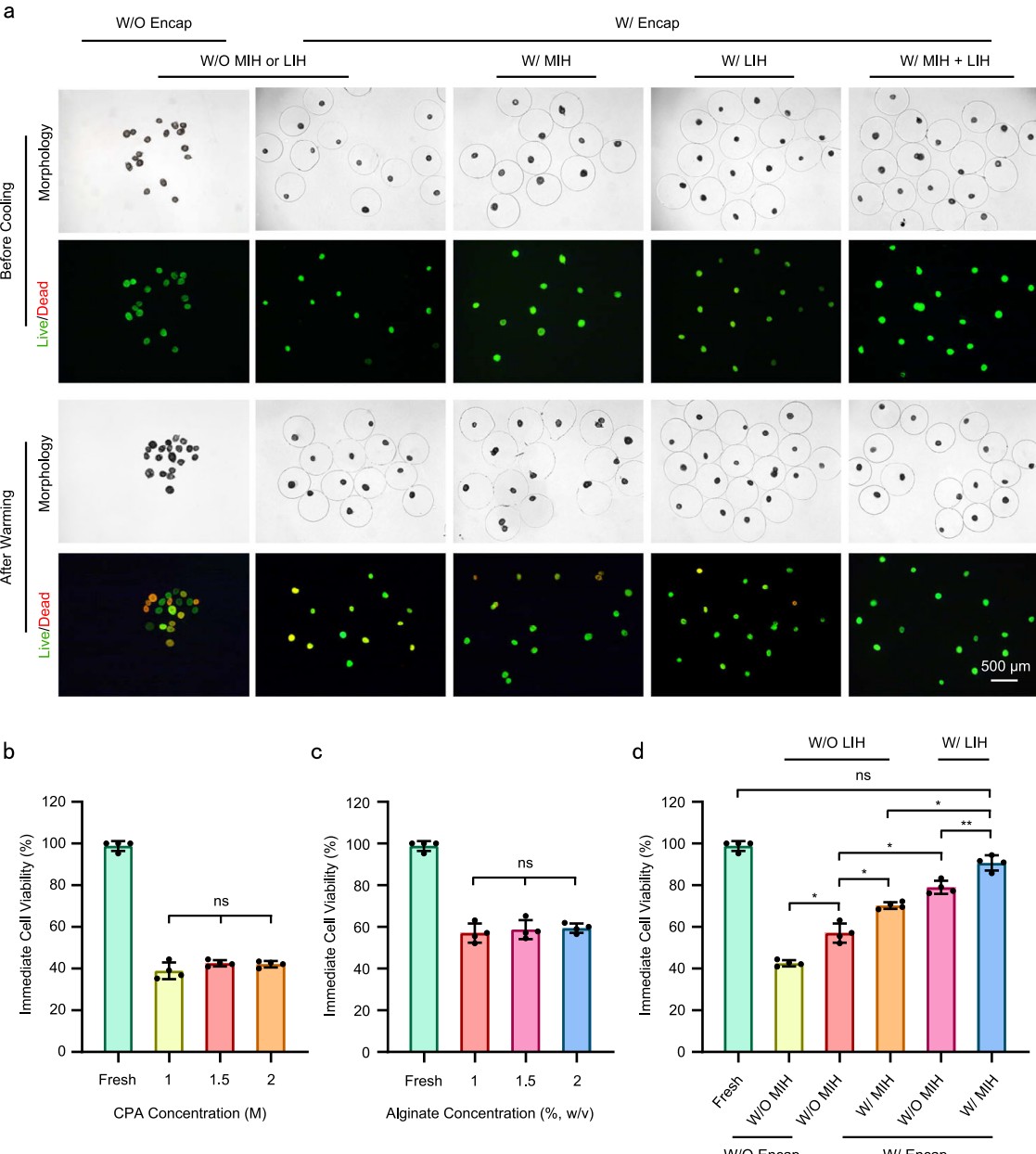

**Fig. 3 | Effect of MIH and LIH on PAFs during warming. a** Differential interference contrast (DIC) and fluorescence images of PAFs after warming under the following different conditions: W/ Encap or W/O Encap, W/ LIH, W/ MIH or W/ MIH and LIH during warming. **b** Viability of the PAFs after vitrification with different CPAs (0 M, green; 1 M, light green; 1.5 M, orange; 2 M, purple bars). $P_{1-1.5} = 0.5978$, $P_{1-2} = 0.4151$, $P_{1.5-2} = 0.9785$. Each experiment was repeated three times with similar results, and representative images are shown. **c** Viability of the PAFs encapsulated in different concentrations of alginate hydrogel (0%, green; 1%, rust; 1.5%, pink; and 2%, emerald green bars. w/v) after vitrification with different 1.5 M CPA. $P_{1-1.5} = 0.9553$, $P_{1-2} = 0.7018$, $P_{1.5-2} = 0.9894$. **d** Viability of PAFs after warming under different conditions (Fresh: green; W/O Encap, MIH and LIH: light green; W/ Encap: orange; W/ Encap and MIH: pink; W/ Encap and LIH: emerald green; W/ Encap, MIH and LIH:

blue bars). $P_{\text{Fresh- W/ MIH+LIH}} = 0.1740$, $P_{\text{W/O Enap-W/ Enap, W/O MIH or LIH}} = 0.0265$, $P_{\text{W/ Enap, W/O MIH or LIH-W/ MIH}} = 0.0306$, $P_{\text{W/ Enap, W/O MIH or LIH-W/ MIH}} = 0.0221$, $P_{\text{W/ MIH-W/ MIH+ LIH}} = 0.0137$, $P_{\text{W/ LIH-W/ MIH + LIH}} = 0.0045$. 1 M: 0.5 M EG + 0.5 M PROH + 1 M trehalose; 1.5 M: 0.75 M EG + 0.75 M PROH + 1 M trehalose; 2 M: 1 M EG + 1 M PROH + 1 M trehalose. W/O Encap: without encapsulation; W/ Encap: with encapsulation; W/ MIH: with magnetic induction heating; W/O MIH: without magnetic induction heating; W/ LIH: with laser-induced heating; W/O LIH: without laser-induced heating. W/ MIH + LIH: with magnetic induction heating and laser-induced heating. One-way analysis of variance (ANOVA) and Tukey's post hoc were used for statistical analysis. ns: $p > 0.05$; *$p \le 0.05$; **$p \le 0.01$. $n = 18$–23 for four replicates. n: number of follicles used in each experiment. Data are presented as the mean ± SD (**b**, **c**, **d**).

0.03%, 0.4% and 0.03%, w/v). Results showed that 0.3% Fe₃O₄ was the most effective (Supplementary Fig. 5a). Besides, GO concentration and laser intensity also affected the survival rate of PAFs. When keeping 0.3% Fe₃O₄ unchanged, the best result was obtained with 0.03% GO, compared with 0.02% and 0.04% GO (Supplementary Fig. 5c). Although there was no statistical significance among the different current and laser intensities, we found that the value of PAF survival rate was slightly higher at 15 A and 3 W/cm² than other groups

(Supplementary Fig. 5b, d). Therefore, PAFs vitrified using 1.5 M CPA with NPs (0.3% Fe₃O₄ and 0.03% GO, w/v) and warmed under 15 A and 3 W/cm² were used for the following experiment.

To detect the impact of vitrification on PAFs, we tested the morphology of the PAFs (Supplementary Fig. 6a, b). The normal vitrified PAFs had a central round oocyte, intact granulosa cell layer and basement membrane that were the same as those of fresh PAFs. For abnormal PAFs, the oocyte contracted severely. The normal PAFs were

green, and the abnormal PAFs were red after fluorescent staining. Besides, the normal vitrified PAFs had normal histological morphology of oocytes and entire granulosa cells (Supplementary Fig. 6c). Growth differentiation factor 9 (Gdf9), fibroblast growth factor 8 (Fgf8) and bone morphogenetic protein 15 (Bmp15), which are specifically expressed in oocytes. They play important regulatory functions in the early development of follicles and the maintenance of granulosa and theca cell functions[45–47]. So, the expression of Gdf9, Fgf8, and Bmp15 was detected (Supplementary Fig. 6d). There was no significant difference on the expression of the Gdf9, Fgf8 and Bmp15 between W/ MIH + LIH and Fresh groups. In summary, the mRNA expression of the three oocyte genes in vitrified PAFs was similar to that in fresh PAFs.

### Evaluation of quality and function of PAFs after vitrification cryopreservation

After warming, in vitro culture of PAFs was performed to detect the effect of vitrification cryopreservation on the performance of the PAFs. The bright field images of PAF development showed that the spherical sphere of the PAFs was well maintained, and an obvious antral cavity appeared on the 10th day, similar to that of the W/O VTF group (Fig. 4a). However, the 3D shape of the follicles could not be retained, and the granulosa cells diffused to the surroundings when fresh PAFs W/O microencapsulation were directly cultured in a 2D environment (Supplementary Fig. 7). Therefore, microencapsulation of PAFs promotes the culture of PAFs in vitro. The assessment of growth of the PAFs post-vitrification is shown in Fig. 4b. The results demonstrated that the diameter (~280 µm) of PAFs vitrified and warmed by MIH + LIH was slightly lower than that of W/O VTF group (~320 µm) on the 13th day. Then, the secretion of estradiol (E2) and progesterone (P4) in culture media which is crucial for the development of PAFs, was further analyzed (Fig. 4c and Supplementary Fig. 8). The result showed that the secretion of E2 from W/ MIH + LIH group was lower than that of the W/O VTF group. There was no significant difference in E2 and P4 secretion between W/O VTF and W/ MIH + LIH groups on days 1 and 8. The percentage of antral follicles post-vitrification (13%) was also lower than that of the W/O VTF group (17%) (Fig. 4e). The pictures of COCs obtained from cultured PAFs were shown in the Fig. 4d. It could be seen that COCs was normal. To further evaluate the quality of antral follicles, we obtained oocytes at the MII stage after culture of PAFs in vitro. The recovery rates of MII oocytes from the vitrified PAFs were comparable to those of the oocytes in the W/O VTF and the fresh oocyte groups (Fig. 4f). A spindle with normal function and structure is essential for chromosome arrangement and separation during oocyte meiosis[48,49]. Spindle morphology and chromosome alignment were assessed by immunofluorescence staining, as shown in Fig. 4g. The barrel-shaped bipolar spindles and the closely arranged chromosomes on the metaphase plate of oocytes (MI and MII) from vitrified PAFs were similar to those of the W/O VTF and fresh oocyte groups, indicating that vitrification did not affect the meiotic division of oocytes (Fig. 4g). In addition, fluorescence images of abnormal spindles are captured in Supplementary Fig. 9. Moreover, the normal spindle rate of oocytes is illustrated in Fig. 4h. Taken together, the results above indicate that the proliferative performance of PAFs and meiotic division of oocytes from PAFs could remain normal after nanowarming with MIH and LIH.

Functional mitochondria are essential for normal oogenesis and embryogenesis[50]. The levels of mitochondrial membrane potential (MMP), adenosine triphosphate (ATP), mitochondrial superoxide (MS) and reactive oxygen species (ROS) are commonly used to assess mitochondrial function[51]. Therefore, the levels of all the indexes mentioned above were measured to assess mitochondrial function in this work.

Normally, in mitochondria, the MMP drives adenosine diphosphate (ADP) into ATP to produce energy under the action of

respiratory chain enzymes[52]. If the MMP value in oocytes is reduced, it may cause a lack of energy which will affect the development of oocytes[53]. Hence, the MMP of MII oocytes was detected by JC-1 staining in this study. The MMP level was measured by the ratio of the red/green fluorescence of the oocytes; the lower the ratio was, the more severe the damage to the MMP was. Typical fluorescence images of the MMP of oocytes are shown in Supplementary Fig. 10a. Analysis of fluorescence value indicated that the red/green fluorescence ratio of the oocytes from the vitrified PAFs was similar to that of the oocytes from the W/O VTF and fresh oocyte groups (Supplementary Fig. 10b). This result suggests that oocytes recovered from the vitrified PAFs still had a good ability to generate energy. To further confirm the energy-producing performance of oocytes, we evaluated the level of ATP in oocytes by a microplate reader. As expected, the ATP value in the MII oocytes from W/ MIH + LIH was lower than the MII oocytes from fresh oocyte group (Supplementary Fig. 10c). MS and ROS are produced by oxidative phosphorylation[51,54]. However, excessive production of MS and ROS will cause excessive oxidative stress, which can lead to damage to the mitochondrial function and the development of cells[51,54]. Typical images of oocytes stained with mitochondrial superoxide indicator (MitoSOX) and 2′, 7′-dichlorodihydrofluorescein diacetate (DCFH-DA) are shown in Supplementary Fig. 10d, e. There were no statistically significant differences in MS and ROS levels in oocytes among the Fresh, W/O VIF and W/ MIH + LIH groups (Supplementary Fig. 10g, h). Intracellular calcium levels are another indicator of mitochondrial function. Both MMP and calcium storage are involved in maintaining cellular homeostasis[55].Therefore, the level of calcium was assessed by fluo-4 staining, and the fluorescence value was analyzed by a microplate reader (Supplementary Fig. 10f, i). There was also no significant difference among Fresh, W/O VIF and W/ MIH + LIH groups, besides, the fluorescence intensity of oocyte from the vitrified PAFs was slight lower than that of oocytes from the W/O VTF and fresh oocyte groups (Supplementary Fig. 10i). The result suggested that vitrification cryopreservation of PAFs had no influence on the concentration of calcium in oocytes. All these results further indicate that the mitochondrial function in oocytes from PAFs post-vitrification is normal.

### Analysis of the developmental capacity of oocytes

To examine the developmental potential of MII oocytes, IVF needs to be performed. Images of mature oocytes and embryos (from the 2-pronuclei stage to the blastocyst stage) are shown in Fig. 5a. There was no difference in embryo morphology among oocytes from the vitrification, the W/O VTF and fresh oocyte groups. Moreover, the development of embryos in each stage (from the 2-pronuclei stage to the blastocyst stage) from the vitrified group was similar to that of the W/O VTF and fresh oocyte groups (Fig. 5b, c). Therefore, the IVF of oocytes and the in vitro developmental performance of embryos were not affected after nanowarming with MIH and LIH.

### Epigenetic analysis of embryos at different stages

As one of the most important epigenetic indicators, histone modifications are of great significance in controlling gene expression and regulating early embryonic development[56]. H3K9me3, H3K4me3 and H3K27ac are typical histone modifications[57–59], so we evaluated them by fluorescent staining (Fig. 6 and Supplementary Fig. 11). The results showed that H3K9me3, H3K4me3 and H3K27ac were similar in W/ MIH + LIH and W/O VTF (Fig. 6 and Supplementary Fig. 11). Among them, H3K9me3 on the paternal genome was erased at the time of fertilization and could be gradually restored afterward. Therefore, H3K9me3 from the paternal parent was not seen during the two-pronuclei stage and could only be observed after the two-cell embryonic stage.

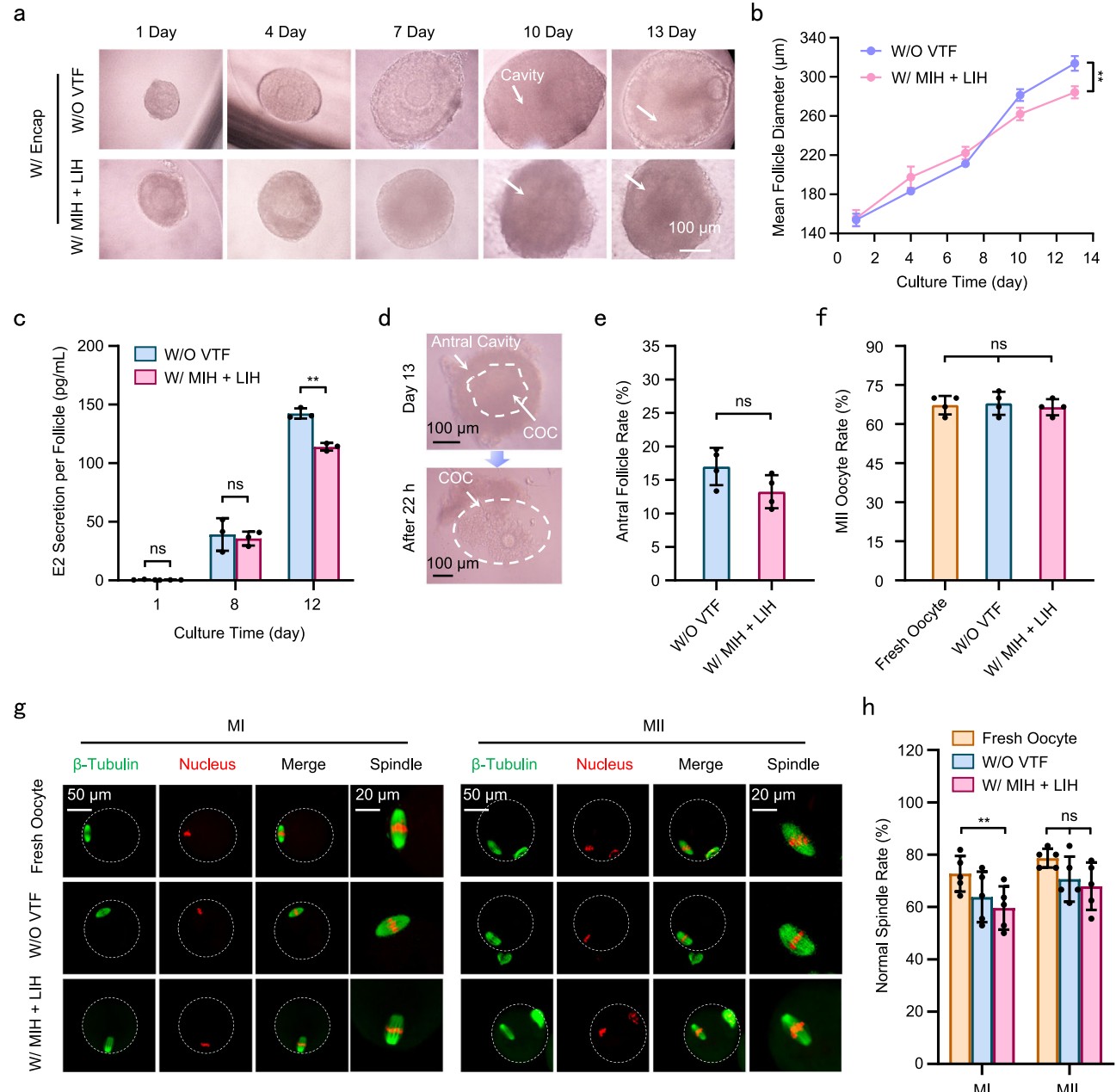

**Fig. 4 | The effect of cryopreservation on the development capacity of PAFs.**
**a** Typical developmental images of PAFs after vitrification (VTF). **b** Growth of PAFs warmed with MIH and LIH. Day 13: $P_{W/O\ VTF-W/\ MIH+LIH} = 0.0011$. **c** Secretion of estradiol (E2) from growing PAFs. Day 12: $P_{W/O\ VTF-W/\ MIH+LIH} = 0.0020$. **d** The pictures of COCs collection. **e** The percentage of PAFs developing to antral follicles. **f** Percentage of MII oocytes in the collected COCs. **g** Confocal images of spindle morphology in oocytes. **h** Normal ratio of spindles in oocytes. MI: $P_{Fresh\ oocyte-W/\ MIH+LIH} = 0.0385$. W/O VTF: PAFs were only encapsulated in hydrogel and did not undergo vitrification; W/ MIH + LIH: PAFs encapsulated in hydrogel were warmed with MIH and LIH after vitrification. Fresh oocytes: MII oocytes obtained directly from the ampulla of the oviduct of a mouse; Two-way analysis of variance (ANOVA) and Tukey's post hoc were used for statistical analysis (**b**, **c**, **h**); One-way analysis of variance (ANOVA) and Tukey's post hoc were used for statistical analysis (**f**); Two-tailed Student's t test (**e**). ns: $p > 0.05$; *$p \le 0.05$. **$p \le 0.01$. $n = 10–20$ for three replicates (**b**); n = 30 for three replicates (**c**); $n = 40–50$ for four replicates (**e**); $n = 10–20$ for four replicates (**f**); $n = 10–20$ for five replicates (**h**). n: number of follicles used in each experiment. Data are presented as the mean ± SD (**b**, **c**, **e**, **f**, **h**). Each experiment was repeated three times with similar results, and representative images are shown (**a**, **d**, **g**).

## Birth of mouse pups after transplantation of 2-cell stage zygotes

To further confirm the development of zygotes in vivo, we carried out embryo transfer experiments. The 2-cell embryos were transferred into the oviducts of surrogate mice (Supplementary Fig. 12). Pictures of newborn mice and 2-week-old mice are shown in Fig. 7a. The mice from the vitrified group grew well. The birth rate was evaluated (Fig. 7b). The result suggests that there was no visible difference among groups of fresh oocytes, W/O VTF, and W/ MIH + LIH. Moreover, the

development of embryos in vivo from the vitrified group was similar to that of the W/O VTF and fresh oocyte groups (Fig. 7c). To study the development of the first-generation mice, we further mated the first-generation mice naturally, and the second-generation mice were successfully born (Fig. 7d). The developmental capacity of the second generation of mice is exhibited in Fig. 7e. All the above results indicate that the developmental potential of embryos is not influenced by vitrification cryopreservation.

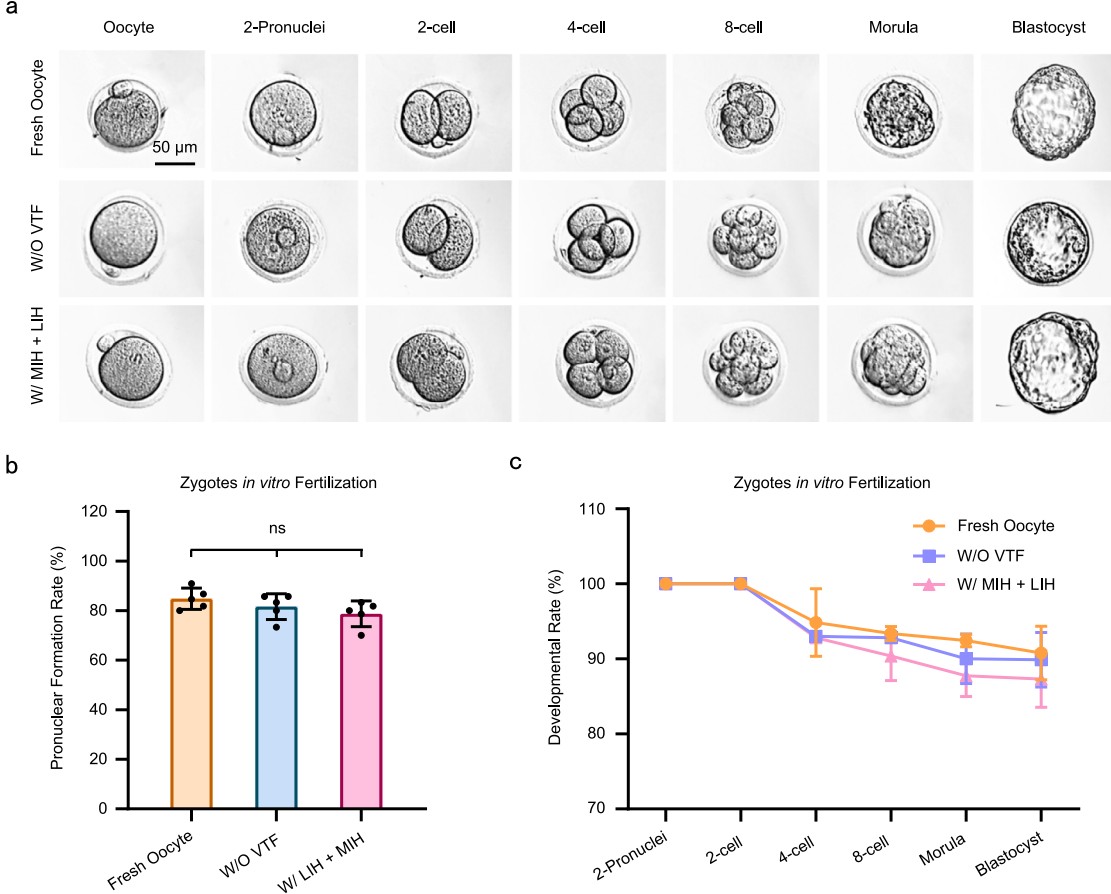

**Fig. 5 | IVF of MII oocytes and development of zygotes. a** Representative images of the development of oocytes in vitro (from MII oocytes to the blastocyst stage). Each experiment was repeated three times with similar results, and representative images are shown. **b** IVF analysis of oocytes. Pronuclear formation rate: Percentage of 2-pronuclei to total number of oocytes involved in IVF. **c** Evaluation of the embryonic developmental performance. W/O VTF: PAFs were only encapsulated in hydrogel and did not undergo vitrification; W/ MIH + LIH: PAFs encapsulated in hydrogel were warmed with MIH and LIH after vitrification; Fresh oocytes: MII oocytes obtained directly from the ampulla of the oviduct of a mouse; One-way analysis of variance (ANOVA) and Tukey's post hoc were used for statistical analysis (**b**); Two-way analysis of variance (ANOVA) and Tukey's post hoc were used for statistical analysis (**c**). ns: $p > 0.05$. $n = 10-20$ for five replicates (**b**); $n = 10-20$ for three replicates (**c**). n: number of follicles used in each experiment. Data are presented as the mean ± SD (**b**, **c**).

## Mechanism of microencapsulation and nanowarming to improve the efficiency of low-pCPA PAF vitrification

The rationale for nanowarming to improve PAF survival is presented in Fig. 8. When the unencapsulated PAFs are vitrified with low-pCPAs, obvious devitrification/recrystallization appears during warming. This phenomenon will trigger the formation of intracellular ice, which will cause severe damage to PAFs. Therefore, the oocyte will be severely shrunken and many granulosa cells will die after traditional water bath warming. Although alginate hydrogels can minimize ice crystal formation during cooling and warming, the result remains unsatisfactory due to slight intracellular ice formation. Shrinkage of oocytes and death of granulosa cells still occurred after warming. MIH was then applied to enhance the rate of warming and thus reduce the nucleation and growth of ice crystals. The oocytes shrunk slightly, and only a small number of granulated cells died after vitrification. However, the magnetothermal effect did not completely eliminate cryoinjury. Similarly, the photothermal effect was not sufficient to avoid PAF damage that may result from a limited heating rate. Therefore, the encapsulated PAFs were warmed with MIH and LIH together after vitrification, which may substantially improve the warming rate. In this way, PAFs can quickly pass through the dangerous temperature zone, reducing the damage of intracellular ice to PAFs. Consequently, the combination of MIH and LIH is an ideal strategy to realize low-pCPA vitrification cryopreservation of PAFs.

## Discussion

Low-pCPA vitrification cryopreservation, which combines the low toxicity of traditional slow cryopreservation with the easy operation of vitrification for preservation, is considered to be an ideal cryopreservation method for the future. However, devitrification/recrystallization during the warming process has always been a problem that urgently need to be solved. Increasing the heating rate was shown to shorten the residence time of PAFs in the dangerous temperature zone, thus minimizing the recrystallization damage of PAFs[39]. Here, we effectively improved the warming rate [Fig. 2a (**iii** and **iv**)] by applying the nanowarming technology combining MIH and LIH, reducing the time of devitrification/recrystallization [Fig. 2a (**v**)]. In addition, compared with high-CPA vitrification and water bath warming reported in previous literature[60], the low-CPA vitrification and nano-rewarming can effectively improve the post-freeze survival rate of PAFs.

In a previous study, we enabled low-cryoprotectant vitrification of stem cell-laden nanocomposite hydrogels by a $GO-Fe_3O_4$ nanocomposite alginate hydrogel ice suppression platform[36]. However, multifunctional photo- and magneto-responsive $GO-Fe_3O_4$ nanocomposites require additional synthesis, which is a complex process. Moreover, after synthesis, the ratio of the two materials (GO and $Fe_3O_4$) cannot be arbitrarily adjusted. Here, we directly introduced two kinds of NPs with photothermal and magnetothermal properties,

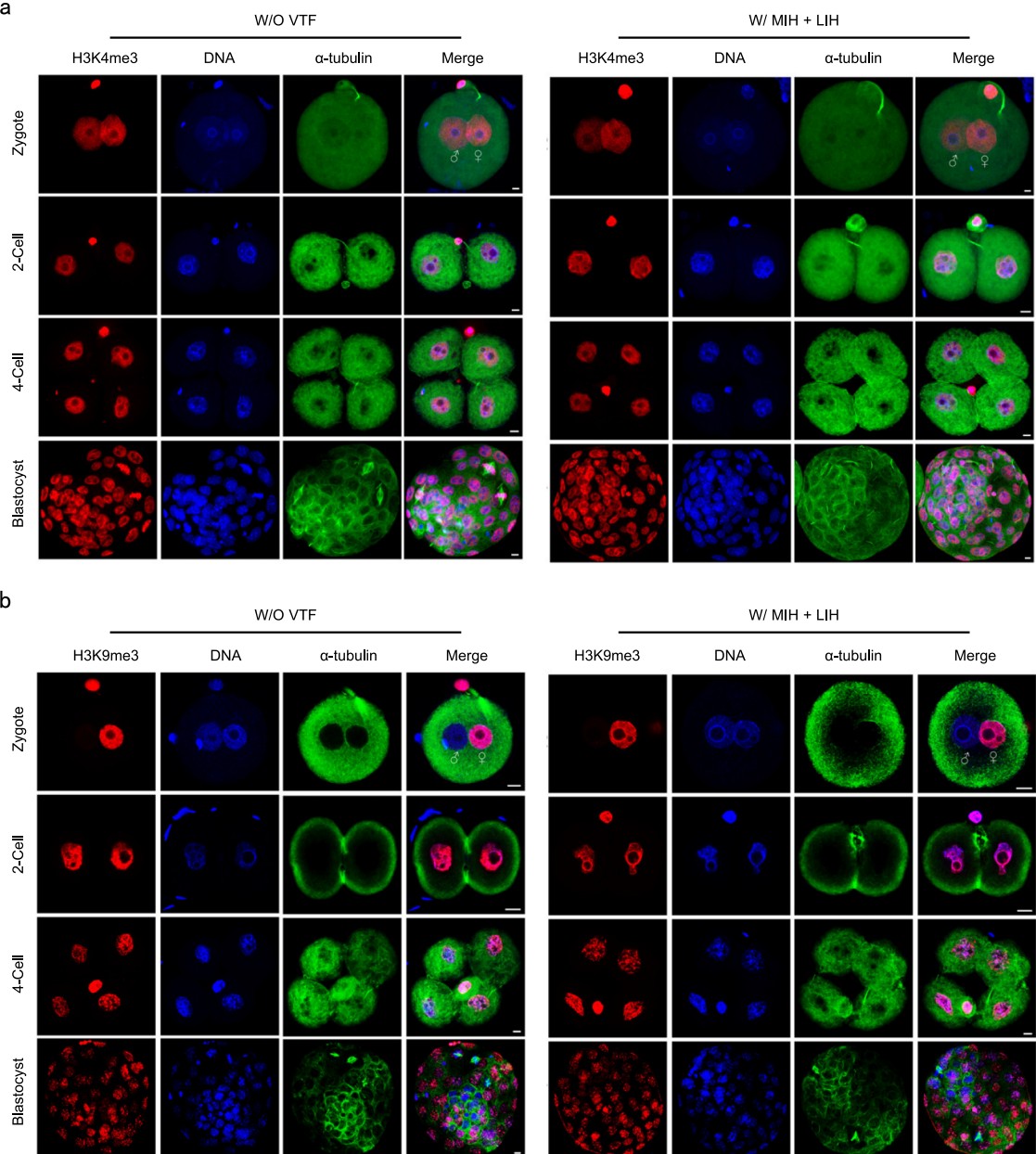

**Fig. 6 | H3K4me3 and H3K9me3 in embryos at different stages. a** Representative images of H3K4me3 in embryos at different stages. **b** Representative images of H3K9me3 in embryos at different stages. Scale bar is 10 μm (**a**, **b**). Each experiment was repeated three times with similar results, and representative images are shown (**a**, **b**). W/O VTF: PAFs were only encapsulated in hydrogel and did not undergo vitrification; W/ MIH + LIH: PAFs encapsulated in hydrogel were warmed with MIH and LIH after vitrification.

without any other synthesis or pretreatment steps. Moreover, the concentration and recipe of the two NPs can be arbitrarily adjusted to obtain our desired warming rate.

From the current point of view, the introduction of NPs into cryopreservation systems in physical field-assisted space heating is inevitable. However, as exogenous and non-natural NPs, their biocompatibility is a problem that cannot be ignored. Moreover, FDA in the United States has considered the toxic effects of NPs and believes that they are not completely safe for human use[61]. Recent studies have shown that various NPs have some potential toxic effects on the female reproductive system, affecting oogenesis and embryonic development[62,63]. Here, we used hydrogel microencapsulation technology to isolate NPs from cells, eliminating their potential toxicity to PAFs. Of course, in addition to the isolation advantage, hydrogel microencapsulation could also prevent ice crystals from spreading to

the interior during the freezing process, thereby protecting PAFs from extracellular ice damage.

Hydrogel microencapsulation was originally designed to enable safe and efficient cryopreservation of PAFs, but we found that this microencapsulation (with correct mechanical property design) could also be directly used for subsequent 3D in vitro culture. The hydrogel microencapsulation provided a 3D microenvironment similar to the extracellular matrix, which is very conducive to the culture of mouse PAFs[64]. To the best of our knowledge, some studies have developed a biomimetic core-shell platform and realized the proliferation and development of stem cells and PAFs[24,65]. Nevertheless, the fabrication of core-shell microcapsules usually requires comparatively complex devices (such as flow-focusing microfluidic devices[65] or tube-in-tube capillary microfluidic devices[24]), which may unavoidably lead to the loss of cells. Moreover, when encapsulating rare cells (e.g., follicles),

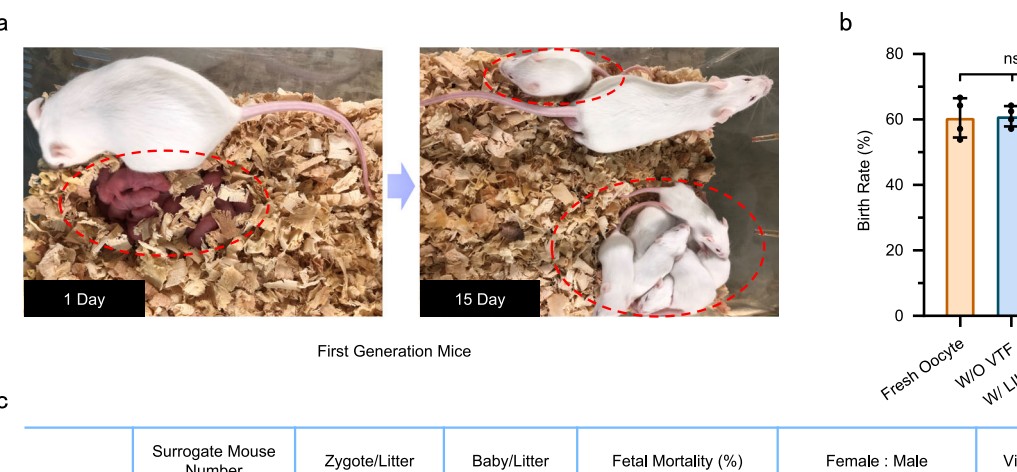

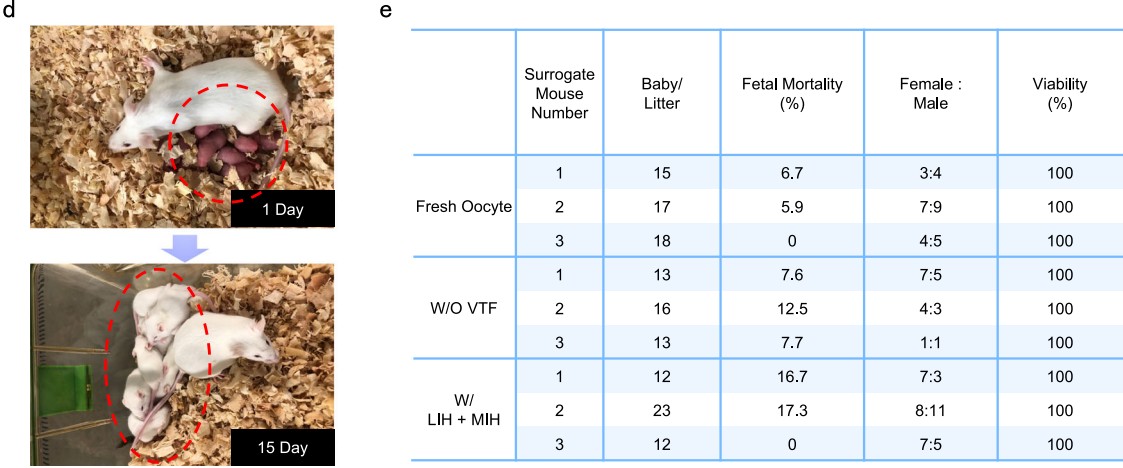

**c**

| | Surrogate Mouse Number | Zygote/Litter | Baby/Litter | Fetal Mortality (%) | Female : Male | Viability (%) |
|---|---|---|---|---|---|---|
| Fresh Oocyte | 1 | 14 | 9 | 35.71 | 5:4 | 100 |
| | 2 | 13 | 7 | 46.15 | 4:3 | 100 |
| | 3 | 14 | 8 | 42.86 | 3:5 | 100 |
| | 4 | 15 | 10 | 33.33 | 7:3 | 90 |
| W/O VTF | 1 | 14 | 9 | 35.71 | 4:5 | 100 |
| | 2 | 16 | 10 | 37.50 | 1:1 | 100 |
| | 3 | 15 | 9 | 40.00 | 7:2 | 100 |
| | 4 | 14 | 8 | 42.86 | 3:1 | 100 |
| W/ LIH + MIH | 1 | 13 | 7 | 46.15 | 5:2 | 100 |
| | 2 | 15 | 9 | 40.00 | 7:2 | 100 |
| | 3 | 9 | 3 | 35.71 | 8:1 | 100 |
| | 4 | 13 | 8 | 38.46 | 5:3 | 100 |

**e**

| | Surrogate Mouse Number | Baby/Litter | Fetal Mortality (%) | Female : Male | Viability (%) |
|---|---|---|---|---|---|
| Fresh Oocyte | 1 | 15 | 6.7 | 3:4 | 100 |
| | 2 | 17 | 5.9 | 7:9 | 100 |
| | 3 | 18 | 0 | 4:5 | 100 |
| W/O VTF | 1 | 13 | 7.6 | 7:5 | 100 |
| | 2 | 16 | 12.5 | 4:3 | 100 |
| | 3 | 13 | 7.7 | 1:1 | 100 |
| W/ LIH + MIH | 1 | 12 | 16.7 | 7:3 | 100 |
| | 2 | 23 | 17.3 | 8:11 | 100 |
| | 3 | 12 | 0 | 7:5 | 100 |

**Fig. 7 | Birth of offspring after transplantation of 2-cell stage zygotes. a** Mouse pups were born after transplantation of 2-cell stage embryos. **b** Analysis of the mouse birth rate. **c** Reproductive performance after zygote transfer. **d** Birth of the second generation of mice. **e** Reproduction performance of the first generation of mice. W/O VTF: PAFs were only encapsulated in hydrogel and did not undergo vitrification; W/ MIH + LIH: PAFs encapsulated in hydrogel were warmed with MIH and LIH after vitrification. Fresh oocytes: MII oocytes obtained directly from the ampulla of the oviduct of a mouse; One-way analysis of variance (ANOVA) and Tukey's post hoc were used for statistical analysis (**b**). ns: $p > 0.05$. $n = 10$–$20$ for four replicates (**b**). n: number of follicles used in each experiment. Data are presented as the mean ± SD (**b**).

many hydrogel microcapsules are empty. Subsequent selective extraction of microcapsules loaded with cells is also required for further applications, which require complex sorting devices[66]. Furthermore, for the better development of cells, it is necessary to precisely control the composition and ratio of the materials in the core or shell[64]. In addition, some reports have shown that alginate hydrogels with low concentrations are suitable for mouse PAF culture[40].

Here, we used a simple centrifugation method to microencapsulate PAFs in alginate hydrogels and found little difference in the survival of PAFs after cryopreservation in the cases of hydrogels with different concentrations. Given the effect of the rigidity and

concentration of hydrogel on the development of mouse PAFs[40], 1% alginate hydrogel was finally selected for the subsequent developmental experiments. After 13 days, hydrogel-encapsulated PAFs developed into preovulation follicles in vitro (Fig. 4a). Fertile oocytes were successfully obtained from these preovulation follicles. Moreover, functional indicators of MII cells were normal, and healthy mouse pups were successfully born after IVF and embryo transfer. Therefore, we developed an all-in-one platform that integrated microencapsulation, cryopreservation and 3D culture potentially for developmental biology, reproductive medicine and toxicology research. Compared with cryopreservation and transplantation of ovarian tissue, this all-in-one

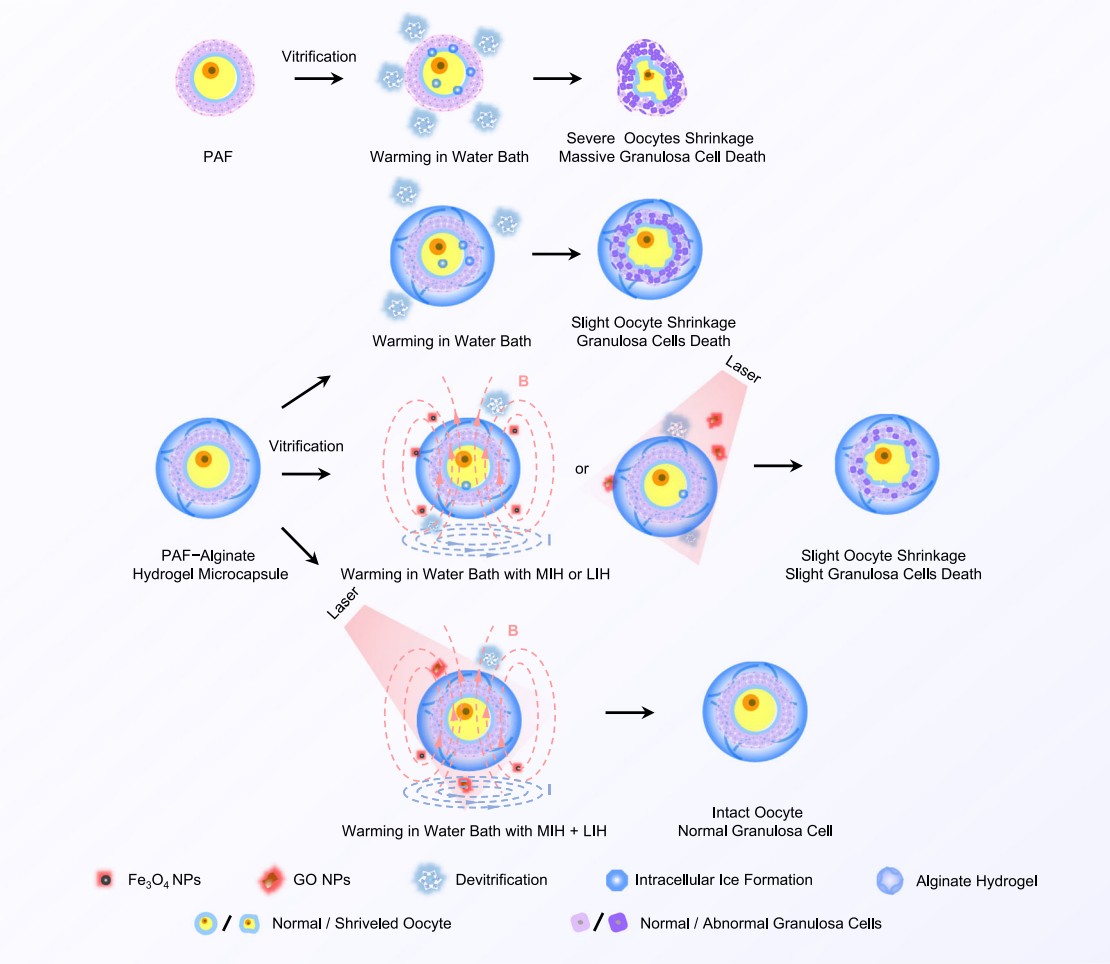

**Fig. 8 | Possible mechanism of microencapsulation, MIH and LIH to improve the efficiency of low-pCPA PAF vitrification.** MIH magnetic induction heating, LIH laser-induced heating.

integrated platform successfully prevented the secondary introduction of malignant cells. Furthermore, hydrogel microencapsulation from the all-in-one integrated platform provides a biomimetic 3D micro-environment for PAF development in vitro, avoiding the surgical trauma of cryopreserved ovarian tissue transplantation in vivo.

Taken together, the combination of nanowarming and hydrogel microencapsulation technology effectively reduced the demand for highly toxic pCPAs, enabling low-pCPA (1.5 M) vitrification cryopreservation of mouse PAFs. PAFs freeze-thawed with this method can develop into preovulatory follicles in vitro. Oocytes obtained from these follicles were then in vitro fertilized and developed into mouse pups after embryo transfer. Therefore, this method provided an integrated platform for microencapsulation, freezing, thawing and 3D culture of PAFs, realizing efficient vitrification cryopreservation and development of PAFs in vitro, which enriched the methods of fertility preservation in women. Adaptation of the protocols for cryopreservation for human ovarian material and PAFs may help improve chances of regaining fertility for patients after treatment. Moreover, our approach can potentially be used to preserve female germ cells of highly endangered animal species, facilitating the creation and development of cryobanks in medical centers worldwide.

## Methods
### Reagents
Three kinds of CPAs were used in this study: 1 M CPA was composed of 0.5 M EG (Sigma-Aldrich), 0.5 M PROH (Sigma-Aldrich) and 1 M trehalose (Sinozyme, China); 1.5 M CPA consisted of 0.75 M EG, 0.75 M

PROH and 1 M trehalose; and 2 M CPA was composed of 1 M EG, 1 M PROH and 1 M trehalose. All CPA solutions were prepared by α-minimum essential medium (α-MEM)-GlutaMAX medium (32571036, Gibco) with 20% (v/v) foetal bovine serum (FBS; S711-001S, Lonsera).

### Animals
Three-week-old female and male KM mice were purchased (Vital River, China). All operations and research were performed in accordance with the protocol of the animal ethics committee of the University of Science and Technology of China. The Animal Ethics number was USTCACUC1801045.

### Isolation of PAFs
PAFs (100–190 μm) were isolated from the ovaries of 3-week-old KM mice by mechanical separation. First, the female KM mice were sacrificed via cervical dislocation. The ovaries were placed in L-15 medium (11415064, Gibco) with 10% (v/v) FBS and 1% (v/v) penicillin-streptomycin (SV30010, HyClone) at 37 °C. Then, PAFs were immediately separated from ovaries using two syringe needles (0.45 × 16 RWLB) and moved to the prepared incubation medium for microencapsulation. The medium was composed of α-MEM-GlutaMAX medium supplemented with 20% (v/v) FBS and 1% (v/v) penicillin-streptomycin at 37 °C.

### Microencapsulation and characterization of PAFs
The microencapsulation of PAFs was performed by a small centrifugal microfluidic device. The centrifugal microfluidic device consisted of a

1.5 mL centrifuge tube and a nozzle. A nozzle (340 μm) was fixed on the tube by a hole made on the cap of the tube. The device was washed by alcohol [conc. 75% (v/v), LIRCON, China] and phosphate buffer solution (PBS; C10010500BT, Gibco) and then exposed to ultraviolet light for 30 min for subsequent use. CaCl₂ solution (1000581922, Sinopharm, China) and sodium alginate (S100127, Aladdin, China) solution were filtered through 0.22 μm pore size filters (L104688, D&B, China) before use. First, 250 μL of 0.15 M CaCl₂ solution was added to the tube, and approximately 250 μL of 1% (w/v) (or 1.5%, 2%) sodium alginate solution containing isolated PAFs was loaded into the nozzle. Next, the device was fixed in a 50 mL centrifuge tube and centrifuged at $100 \times g$ (or $126 \times g$, $156 \times g$) for 2 min. Then, alginate hydrogel microcapsules loaded with PAFs were produced and collected in the separation medium mentioned above for subsequent operation.

The 1% (w/v) alginate hydrogel microcapsules W/ or W/O cooling-warming were freeze-dried by cryomicroscopy (FDCS196, Linkam, UK). Microcapsules were first cooled to $-40\,°C$ at a cooling rate of $100\,°C/min$, and freeze-dried at $-40\,°C$, and 10 Pa for 4 h. Next, gold was sprayed onto the surface of the microcapsules, and external images were obtained with a scanning electron microscopy (JSM-6390 LA, JEOL, Japan).

### Synthesis and Characterization of Fe₃O₄ and GO NPs

Fe₃O₄ NPs were generated by chemical coprecipitation of FeCl₃·6H₂O and FeCl₂·4H₂O. FeCl₃·6H₂O and FeCl₂·4H₂O were added to DI water at $70\,°C$ at a ratio of 1:2. They were dissolved in water by continuous flow of nitrogen and vigorous stirring. Then, the sample was turned black by reacting with NH₃·H₂O for 120 min. Next, after the Fe₃O₄ NPs were isolated from the solution by magnets, they were washed two times with ethanol or water. Finally, the Fe₃O₄ NPs were placed in an oven at $80\,°C$ for 24 h.

The Fe₃O₄ NP morphologies were detected using TEM (H-7650, Hitachi, Japan). The magnetism of the Fe₃O₄ NPs was tested at room temperature by a vibrating sample magnetometer (VSM; MPMS 3, Quantum Design). Moreover, the composition of the NPs was characterized by XRD (X'-Pert MPD, Nalytical, Netherlands), and the angle range was $2\theta = 10-90°$.

GO NPs were synthesized by a modified Hummers' method with natural graphite powder. H₂SO₄ (70 mL) and NaNO₃ (1.5 g) were mixed with graphite powder (3.0 g) in an ice bath using magnetic stirring. Next KMnO₄ (9.0 g) was added to the solution at less than $20\,°C$ and then kept in a water bath at $34\,°C$ for 24 h. Then, DI (590 mL) was added to the solution under stirring, and 30% (w/v) H₂O₂ (15 mL) was added slowly. Subsequently, the slurry was washed several times with DI water, and a neutral pH was acquired. Finally, the slurry was dialyzed and purified for 1 week.

The morphology of the GO NPs was analyzed by TEM. In addition, FTIR (Nicolet 8700, Thermo Scientific) of GO NPs was characterized in the range of $400-4000\,cm^{-1}$. Moreover, Raman spectra were obtained at an excitation wavelength of 514.54 nm by a confocal microprobe Raman system (LabRAM HR Evolution, HORIBA Jobin Yvon, France).

### Toxicity analysis of hydrogel microencapsulation and NPs (Fe₃O₄ and GO) to PAFs

The PAFs with (W/) and without (W/O) encapsulation were incubated in basic medium at $37\,°C$, respectively, to examine the effect of hydrogel microencapsulation on PAFs. The basic medium used for culture consisted of MEM-GlutaMAX medium supplemented with 10% (v/v) FBS and 5 g/mL insulin, 5 g/mL transferrin, 5 ng/mL selenium (ITS; 51500-056, Gibco), and 100 mIU/mL recombinant human follicle stimulating hormone (FSH; NSFH, China). Further to detect the toxicity of NPs (Fe₃O₄ and GO) to PAFs, 0.3% (w/v) Fe₃O₄ and 0.03% (w/v) GO were added to the basal medium to incubate PAFs W/ and W/O encapsulation. Then, the viability of PAFs incubated for different time periods (0, 6, 12, 24, 36, 48, 60 h) was assessed by fluorescent staining. The PAFs

were stained by an acridine orange/ethidium bromide (AO/EB) staining kit (KGA502, KeyGen Biotech, China), and all procedures followed the protocol. Green represents live and red indicates dead when we observed the PAFs under a fluorescence microscopy (Eclipse Ti-U, Nikon, Japan).

### Scanning electron microscopy (SEM) of hydrogel microcapsules and transmission electron microscopy (TEM) of preantral follicles

Hydrogel microcapsules were cultured in two 1.5 mL EP tube by the double distilled water with Fe₃O₄ NPs (0.3%, w/v) and GO NPs (0.03%, w/v) for 24 h at room temperature. Then, one EP tube was transferred to a vial and stored in a $-80\,°C$ refrigerator for 24 h. Fe₃O₄ NPs in another EP tube were sucked to the tube wall by a magnet and then sucked away by a pipette. The microspheres were washed with double distilled water until no NPs were visible and then were put in a vial in $-80\,°C$ refrigerator for 24 h. Next, the two vials were quickly transferred to a vacuum-freeze dryer (FDU-2110, EYELA, Japan) and lyophilized at $-80\,°C$. Freeze-dried microspheres were sputtered with gold, and images of their interior and exterior were obtained by scanning electron microscopy (SEM; JSM-6390 LA, JEOL, Japan).

To detect whether NPs were took in by PAFs, the PAFs encapsulated in hydrogel microscopes were incubated within Fe₃O₄ NPs (0.3%, w/v) and GO NPs (0.03%, w/v) for 24 h at $37\,°C$. Then the PAFs were released from hydrogel microscopes and collected. Afterwards, the PAFs were fixed with 3% glutaraldehyde at $4\,°C$ overnight for subsequent analysis. Next day, the PAFs were incubated 0.1 M potassium ferrocyanide and 1% (w/v) osmium tetroxide solution for 40 min after washing by 0.1 M cacodylate buffer for three times. Then, the PAFs were incubated by 1% (w/v) osmium tetroxide solution for 40 min again after washing three times by cacodylate buffer. Next, the PAFs were incubated by 2% (w/v) uranyl acetate for 1 h after washing by cacodylate buffer and double distilled water for three times. The PAFs were dehydrated by different concentrations of alcohol (30%, 50%, 70%, 80%, 95%, 100%, 100%) for 15 min. Then, the PAFs were incubated in 25%, 50%, 100%, and 100% epon for 3, 5, 12, and 24 h. After drying, it was sliced through a microtome (UC7, Leica, Germany). And pictures were captured by a transmission electron microscope (TEM; H-7650, Hitachi, Japan).

### Magnetothermal and photothermal systems

The MIH system (SPG-10AB-II, Shuangping, China) was composed of a water tank, a water pump, a $37\,°C$ water bath and a magnetothermal device with an intermediate frequency generator and a water-cooled copper coil (2 cm in diameter). The LIH system (LSR808H, Lasever, China) consisted of an 808 nm wavelength fiber connected to a controller and an aluminum stand.

### Vitrification Cryopreservation of Microcapsules Containing PAFs

In this work, the CPA solutions consisted of pCPAs (0.75 M EG and 0.75 M PROH) and nonpenetrating CPAs (1 M trehalose). The CPA was used with different concentrations of Fe₃O₄ NPs (0.2%, 0.3% and 0.4%, w/v) and GO NPs (0.02%, 0.03% and 0.04%, w/v). The PAF-alginate microcapsules were immersed in CPA and equilibrated at $4\,°C$ for 15 min. Then, the CPA and microcapsules were loaded into a PS (ST025, FHK, Japan). Next, the PS was plunged into LN₂ for 30 min.

### Warming by MIH and LIH

For the CPA without NPs, the PS without NPs was quickly placed in a $37\,°C$ water bath and shaken rapidly. Then, the CPA solution was removed by centrifugation at 1000 rpm after warming. For the MIH group, PS was quickly placed in a $37\,°C$ water bath in a coil device with different currents (5, 15 and 25 A). Next, after warming, Fe₃O₄ NPs were removed by a magnet and the CPA solution was removed as above.

For the LIH group, the PS was quickly transferred into a water bath at 37 °C under the laser spot (diameter was approximately 5 mm). Laser with different powers (3, 4 and 5 W/cm$^2$) were kept approximately 5 cm away from the PS. Then, the microcapsules were moved into a dish containing 3 mL α-MEM supplemented with 10% (v/v) FBS and picked out. After washing three times by α-MEM medium, the CPA solution was removed. And then the microcapsules were kept in separation medium at 37 °C for subsequent experiments. For the combination of MIH and LIH, the PS was transferred into a water bath at 37 °C in a coil device under the laser spot. After warming, the microcapsules were collected following the method mentioned above.

### Thermal history during warming
The transient temperatures of CPA for MIH and LIH were examined using an optical fiber temperature sensor (FOTS-DINA-5040-N, Indigo, China) during cooling and warming.

The effects of MIH and LIH were detected by a thermal imager (Ti25, Fluke). CPA solutions with different concentrations of $Fe_3O_4$ NPs (0.2%, 0.3% and 0.4%, w/v) and GO NPs (0.02%, 0.03% and 0.04%, w/v) were loaded in a 1.5 mL centrifuge tube. The initial temperature of the sample was 36.6 °C. Then the sample was placed in the coil for 45 s or under the laser spot for 45 s. Infrared thermograms with the maximum temperature of each sample were reported by a thermal imager.

### PAF viability
PAF viability post-vitrification was detected by an AO/EB staining kit as mentioned above. The microcapsules were resuspended in 60 μL of separation medium and 3 μL of fluorescent dye solution and incubated for 3 min. Then, fluorescence images of PAFs were observed under a fluorescence microscopy (Eclipse Ti-U, Nikon, Japan).

### In vitro culture of PAFs after vitrification cryopreservation
The encapsulated PAFs were cultured in 96-well plates with 200 μL of culture medium (one microcapsule in each well). The culture medium consisted of α-MEM-GlutaMAX medium supplemented with 20% (v/v) FBS, 1 × ITS, 100 mIU/mL FSH and 1% (v/v) penicillin-streptomycin. The 96-well plate was incubated in a 5% $CO_2$ incubator at 37 °C for 13 days. Then, half of the medium was exchanged with fresh medium every other day.

### Analysis of PAF diameter and secretion of estrogen and progesterone
The morphological characteristics and diameter of PAFs were analyzed by inverted microscopy (BX-53, Olympus, Japan) after 1, 4, 7, 10 and 13 days of culture. The diameter was the mean of two cross measurements of each PAF by ImageJ software (1.52a, National Institutes of Health). To detect the level of steroid secretion by PAFs, 20–30 PAFs were cultured in 4 mL medium in a 60 mm dish and half of the medium was replaced by fresh medium every other day. The culture medium of different days (1, 8, 12 days) was collected and stored at −20 °C for subsequent detection of estrogen (E2) and progesterone (P4) secretion levels. The concentrations of estrogen and progesterone in the collected culture medium were analyzed by ELISAs (ELISA kit, Joyee, China). PAFs Colorimetric analysis was performed by a microplate reader (SpectraMax iD5, Molecular Devices) at 450 nm.

### Histology and quantitative RT-PCR
PAFs were fixed in 4% (w/v) paraformaldehyde (PFA; P6148-500G, Sigma-Aldrich) for 2 h at room temperature, and then, the PAFs were embedded in paraffin (411663, Sigma-Aldrich). Next, the paraffin block was sliced at a thickness of 5 μm. Finally, the PAF sections were stained with hematoxylin (MHS32, Sigma-Aldrich) and eosin (230251, Sigma-Aldrich).

The expression of three genes related to folliculogenesis and oogenesis was detected by qRT-PCR. The microcapsules were

dissolved in sodium citrate, and PAFs were released. Then, RNA was extracted by TRIzol (B511311, Sangon, China) and treated with DNase (EN0523, Thermo Scientific). Next, reverse transcription was performed with an iScript cDNA Synthesis Kit (AE301-02, TransGen, China). qRT-PCR was carried out by SYBR Green master mix (Q111-02, Vazyme, China) on a real-time monitor (LightCycler 96, Roche, Switzerland). All the primer sequences are shown in Supplementary Table 1.

### In vitro maturation of antral follicles and acquisition of MII oocytes
On the 13th day of culture, the alginate hydrogel was dissolved in 75 mM sodium citrate solution. Then, the antral follicles were incubated in 200 μL of culture medium [α-MEM-GlutaMAX medium supplemented with 5 μg/mL epidermal growth factor (EGF; 315-09, Peprotech), 2.5 IU/mL human chorionic gonadotropin (HCG; NSFH, China), 20% (v/v) FBS, 1 × ITS, 100 mIU/mL FSH and 1% (v/v) penicillin-streptomycin]] in a 5% $CO_2$ incubator at 37 °C for 48 h. The cumulus-oocyte complex (COC) was obtained by mechanically puncturing mature antral follicles with a syringe needle. Then, the oocytes were released from the COC after incubation in 80 IU/mL hyaluronidase (Solarbio, H8030, China) for 5 min.

### Collection of oocytes
Six- to eight-week-old KM female mice were injected with 10 IU pregnant mare serum gonadotropin (PMSG; NSFH, China) in the abdominal cavity, and then, 10 IU HCG was injected after 48 h. After 14–16 h injection of HCG, the COC was removed from the ampulla of the fallopian tube with a syringe needle. Then, COC was treated with 80 IU/mL hyaluronidase at 37 °C for 5 min and washed 3 times with PBS. The obtained oocytes were transferred to M2 medium (M7167, Sigma-Aldrich) for subsequent experiments.

### Analysis of oocyte spindle and chromosome configuration
The morphology of the oocyte spindles and chromosomes was evaluated by immunofluorescence staining. First, the MII oocytes were fixed in 4% PFA for 30 min at 4 °C. Then, oocytes were permeabilized in 5% Triton X-100 (X-100, Sigma-Aldrich) for 20 min and blocked in 3% (w/v) bovine serum albumin (BSA; B2064, Sigma-Aldrich) for 1 h at room temperature. Next, oocytes were incubated with monoclonal anti-β-tubulin antibody (SAB4200715, Sigma-Aldrich) (1:500 dilution) for 24 h at 4 °C. Then, the oocytes were incubated in Alexafluor 488 rabbit anti-mouse IgG (H + L) secondary antibody (A-11059, Invitrogen) (1:500 dilution) for 1 h at room temperature after washing three times with blocking solution (5 min each time). Then, the DNA was stained with 5.0 μg/mL Hoechst 33342 solution (C0030, Solarbio, China) for 20 min and washed three times with PBS (3 min each time). Finally, images of oocytes were taken by a confocal laser scanning microscope (C2 Plus, Nikon, Japan).

### Evaluation of oocyte MMP, ATP, MS, ROS and calcium ions levels
To test the mitochondrial function of oocytes, oocytes were divided into five major experimental groups to assess the levels of MMP, ATP, MS, ROS or calcium, respectively (each group contained three groups: Fresh, W/O VIT and W/ MIH + LIH). For this, oocytes were incubated in JC-1 staining solution (KGA601, KeyGen, China), CellTiter-Glo Reagent (Luminescent cell viability assay kit, Promega), MitoSOX Red Mitochondrial Superoxide Indicator (M36008, Invitrogen), DCFH-DA (D6470, Solarbio, China) staining solution or Fluo-4 AM (KGAF024, Keygen, China), respectively, for 20–30 min respectively at 37 °C in a 5% $CO_2$ incubator. Then oocytes were placed in 96-well black assay plates (6–8 oocytes/well) (3922, Corning) after washing three times with PBS. Subsequently, the fluorescence/luminescence intensity was measured by a multimode reader (SpectraMax iD5, Molecular Devices). And the fluorescence/luminescence intensity was divided by the

number of oocytes to calculate the relative fluorescence value of each oocyte. Among them, the ratio of red/green fluorescence intensity of each oocyte was used as an indicator to measure MMP. Finally, fluorescence images of oocytes were taken by confocal laser scanning microscopy (LSM 710, Zeiss, Germany).

### IVF, embryo development in vitro and in vivo transfer
First, sperms were obtained from the epididymis of male KM mice at 8 weeks. The sperms were washed with equilibrated washing solution to remove epididymal tissue, and then, the sperms were incubated in a 37 °C and 5% $CO_2$ incubator for 30 min to confirm that the sperms could swim freely. Then, the sperms with good viability were transferred to the equilibrated washing solution and incubated for 2 h. After that, oocytes were incubated with $5 \times 10^4$ sperms for 6 h in fertilization medium (K-SIFM-50, Cook Medical). Subsequently, 2-pronuclei stage embryos were transferred into cleavage medium (K-SICM-50, Cook Medical) and incubated in a 5% $CO_2$ incubator at 37 °C. In the next few days, the different stages of embryo development in vitro were recorded through DIC microscopy.

As pseudopregnant mice, 8-week-old female mice were mated with male mice whose vas deferens had been removed. Then, the ovaries, fallopian tubes and uterus of the mouse were found through routine surgery after anesthesia. An incision was made in the fallopian tube, and embryos (9–16) were injected into the fallopian tube through a microtubule at a time. After the wound was sutured, the mice were transferred to a suitable environment for culture until the mouse pups were born. After 3 weeks, male and female pups were raised in different cages. At the ninth week, a female mouse and a male mouse were placed in a cage to produce the next generation of mice. The birth of the pups was assessed after approximately three weeks.

### Assessment of embryonic histone modifications
The embryonic histone modifications were evaluated by immunofluorescence staining. First, embryos from different stages (2-pronuclei stage, 2-cell, 4-cell and blastocyst) were collected after IVF. Next, the embryos were fixed, permeabilized and stained for tubulin (anti-α-tubulin-FITC, F2168, Sigma-Aldrich), and DNA by the method previously described. Then, the embryos were incubated with Anti-H3K9me3 antibody (A2360, ABclonal, 1:100 dilution), Anti-H3K4me3 antibody (ab8580, Abcam, 1:100 dilution), and Anti-H3K27ac antibody (8173, Cell Signaling Technology, 1:100 dilution) for 24 h at 4 °C. After that, the embryos were incubated in donkey anti-rabbit IgG (H + L) highly cross-adsorbed secondary antibody Alexa Fluor 555 (A-11059, Invitrogen) (1:500 dilution) for 1 h at room temperature after washing three times with blocking solution (5 min each time). Finally, images of oocytes were taken by a confocal laser scanning microscope (C2 Plus, Nikon, Japan).

### Statistical analysis
Experimental data are expressed as the mean ± standard deviation from at least three independent measurements. Statistical significance was analyzed by ANOVA with Tukey's post hoc or two-tailed Student's $t$ test. ns: $p > 0.05$; *$p < 0.05$; **$p < 0.01$; ***$p < 0.001$. Statistics and graphs were obtained using Prism 8 (GraphPad Software). Images were processed by ImageJ (1.52a, National Institutes of Health) and Image-Pro Plus (6.0, Media Cybernetics, Inc.).

### Reporting summary
Further information on research design is available in the Nature Portfolio Reporting Summary linked to this article.

## Data availability
All relevant data supporting the key findings of this study are available within the article and its Supplementary Information files or from the corresponding authors upon reasonable request. Source data are provided with this paper.

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

## Acknowledgements

We acknowledge Dr. Menghan Wang, Dr. Xianhui Qin and Dr. Faryal Farooq Cheepa for the fruitful discussions. In addition, we would like to thank Dr. Xianhong Tong from The First Affiliated Hospital of USTC and Zhiguo Zhang from The First Affiliated Hospital of Anhui Medical University for the supply of important reagents. This work was supported by the National Natural Science Foundation of China (no. 82172114) and the Anhui Provincial Natural Science Foundation for Distinguished Young Scholars (no. 2108085J37).

## Author contributions

G.Z., Q.S. and Y.C. conceived the project; G.Z. and Q.S. supervised the study; C.T., L.S. and C.G. conducted the experiments; C.T., L.S., C.G., Y.C., Q.S. and G.Z. analyzed the data; C.T., L.S. and G.Z. wrote the paper; G.Z. edit the paper; and all authors approved the paper.

## Competing interests

The authors declare no competing interests.
