## [Peer Review File · Nature Communications]

Microencapsulation and Nano-warming Enables Vitrification Cryopreservation of Mouse Preantral FolliclesReviewers' Comments:

Reviewer #1:

Remarks to the Author:

The manuscript shows very interesting and successful results of mouse preantral follicles (PAFs) vitrification associating microencapsulation in alginate and nanowarming using magnetic and light induced hyperthermia by mixing iron oxide and graphene NPs to the vitrification solution. With this approach, the authors present a protocol that allows high survival of vitrified PAFs, which are able to develop in vitro to antral follicles and produce viable oocytes that can be in vitro fertilized and produce live young after embryo transplantation to surrogate mothers. The study is elegant and very complete. Methodology and results are sound, and figures are informative and self-explanatory. The study is well based on pertinent literature.

The study brings a novel methodology that may be of significance to the field of animal and human fertility preservation.

Only a few points must be considered to make the manuscript clearer.

Major points:

Please, change "in situ" to 'in vitro' throughout the text (there are 9 places where the term "in situ" is used wrongly, including the title).

Lines 211-214 - Please, include the duration (in time units) of ice crystals in each of the 4 treatment groups in the text.

Lines 266-267 - Where are the statistics of this result? The figure doesn't show statistical differences among CPA concentration groups.

Lines 290-300 - There are no statistical differences to support those affirmations, at least not shown on the Figure. If there is any statistical significance among the different current /light intensities, please include on the manuscript. I don't see any problem in the choice being random in this case, but it must be clear in the text.

Lines 401-403 - I don't see this difference for MS in the figure (for instance, a similar difference in the bars size is observed in Fig. 8i and you say they were similar in lines 409-410). For ROS the graphic's bar is a little smaller, but there is no statistical difference indicated. The only significant difference on mitochondrial function among Fresh, W/O VIT and W/ MIH+LIH is on ATP level (Fig. 8c). Authors should stick to statistical differences and be consistent throughout the text.

Pg 47-49 – I think the Methods for detection of MMP, ATP, MS, ROS and Ca⁺ levels could be described together, since they all refer to mitochondrial function and because some sentences are repeated several times on those topics (for example: "Then, oocytes were placed in 96-well black assay plates (6 oocytes/well) (3922, Corning) after washing three times with PBS", "the relative fluorescence of each oocyte was calculated", "images of oocytes were obtained by confocal laser scanning microscopy").

Minor points:

Line 56 – Change "fertility in women preservation" to 'fertility preservation in women'

Line 67 – Change "transplanted ovarian tissue" to 'ovarian tissue to be transplanted in the future'

Line 102-103 – The problems pointed out for IRIs apply to NPs as well. Please, delete "almost all IRIs are exogenous, non-natural, non-Food and Drug Administration (FDA) approved additives. Moreover,".

Line 129 – Change "granule/membrane cells" to 'granulosa cells'

Line 205 – Please change "transparent" to translucent

Line 217 - Inhibit or minimize? I think I can still see some ice crystal formation in the pictures and the video.

Line 307 - What do you mean by "membrane cells"? Do you mean basement membrane? There are no cells on the basement membrane. Please, clarify.

Lines 315-316 - Not for Fgf8 - there is a statistical difference appointed on the figure.

Lines 322 and 348 – Change “probability” to ‘percentage’

Line 325/Figure 4 - The nomenclature of the groups is a little confuse. Please, explain on the figure caption what exactly are W/O VTF and W/ LIH+MIH as well.

Line 339 – Change “proliferation” to ‘growth’

Line 340 – Change “of the PAFs” to ‘of the vitrified and warmed by MIH+LIH PAFs’

Lines 355-359 – This statement is true but does not apply to this study since mature oocytes were not cooled/warmed. The spindle analysis is important, but this piece of text is not necessary. Please, delete.

Figure 6 - What do you mean by "Birth sequence"? Please, clarify.

Line 435 – Change “probability” to ‘rate’

Lines 446-447 – Change “the reproductive performance” to ‘development’

Line 450 – I think you meant second generation (F2) instead of third generation. Please, correct.

Line 469 – Change “atrophied” to ‘shrunk’

Line 574 – Include ‘meiotic’ before “spindles”

Line 586 - Include ‘in vitro’ before “fertilized”

Line 620 - Were the PAFs dissected individually? Or was the tissue macerated and PAF harvested afterwards? Please, clarify in the text.

Line 621 - It would be better to mention the Gauge of the used needle.

Line 681 - PAFs' TEM (showed in Supplementary Figure 3) methodology was not described in the text.

Line 759 - Supplementary Table 1 only brings the primer sequences of 3 genes, and the results shown are also for 3 genes only. Please, correct the text accordingly.

Line 783 – Please, change “vertical” to ‘cross’

Supplementary Material:

In general, supplementary material is helpful and well connected to the manuscript. Supplementary Figure 1 and Supplementary Movie 1 are very helpful for understanding the alginate microcapsules formation and structure.

Minor points:

Supplementary Movies 2-5 could be in slow motion to allow a better visualization of the warming process.

Supplementary Figure 6 – The figure says the follicle is fresh, while the figure caption says “PAF-alginate hydrogel microcapsules dissolved in 75 mM sodium citrate” and the text (lines 336-337) says “when the PAFs were not encapsulated by hydrogels”. Was this follicle encapsulated while fresh and then the microcapsule dissolved before placing in IVC, or not encapsulated at all? Please, be more specific and coherent.

Reviewer #2:

Remarks to the Author:

Cryopreservation of PAFs is a promising method for fertility preservation. Although vitrification has potentials to become a superior approach, it still needs further improvement. The authors focused on the thawing step of cryopreserved PAFs by assessing the beneficial effects of the alginate hydrogel microcapsules as well as the nanowarming using NPs. Whilst interesting, there are below concerns need to be addressed.

Major concerns

1. It is overstating to mention about clinical application. Due to large size of preovulatory antral follicle and long duration from PAF to preovulatory antral follicle for development in human, the concept of in vitro culture of PAFs shown in this manuscript is difficult to apply. The hydrogel-microcapsuled PAFs may be able to graft into patient ovaries, the hydrogel microcapsules could prevent vascular

reconstruction to the follicles.

2. Many experiments are lacking the control groups without hydrogel microcapsules. Furthermore, the number of samples in most of experiments is unclear. Although some experiments indicate the sample numbers, they are too small (n=3) to lead the authors' conclusion.

3. Evaluation of epigenetic changes in oocytes/embryos recovered after this approach is missing.

4. No data is shown to prove whether NPs are completely separated from PAFs.

Individual concern

5. The reviewer does not agree that PAFs can be obtained from the ovaries of old women. Indeed, the number of PAFs in old women is very limited and difficult to isolate due to excess fibrosis in ovaries.

6. Figure 2B. The explanation of "****" in the graph is lacking.

7. Figure 2. High magnification images of PAFs in the hydrogel microcapsules need to be included.

8. Supplemental Movie 1. In this movie, PAF locates in the peripheral position of the hydrogel microcapsules. It is unclear if such location of PAF affects the efficiency of nanoworming and following in vitro culture procedures. The movie shows some abnormal shape microcapsules (not perfect sphere) without PAFs. The data is not given whether this abnormal shape microcapsules interferes the outcome of experiments. Based on the movie and materials and methods, multiple PAFs could be encapsulated in one microcapsule and such PAFs might be difficult to develop after thawing and cultures. Together with considering the formation of abnormal shape microcapsules, the success rate of preparation of proper PAFs in the hydrogel microcapsules should be presented.

9. Movies 2-5. These movies are not suitable for evaluating the ice-crystal formation.

10. Supplemental Figure 3. The reviewer could not detect the hydrogel microcapsules in pictures. (b) needs positive controls and another control group without the hydrogel microcapsules. (c) needs explanation of pictures.

11. Supplemental Figure 5. Similar to the Supplemental Figure 3, The reviewer could not detect the hydrogel microcapsules in pictures. Control groups without the hydrogel microcapsules need to be included in the experiments. There is no explanation why the authors measured the transcript levels of oocyte-derived factors on day 13 of cultures.

12. Figure 4. (a) the explanation of arrow is lacking. The data shown in Supplemental Figure 6 need to be shown here. (c and d) The P4 and E2 levels are originate from one follicle based on the materials and methods. If so, they are too high. Also, it seems to be difficult that the PAFs continue to produce P4 and E2 for 12 days without adding androgen substrate. The pictures of COCs obtained from cultured PAFs should be included to demonstrate the normality of cumulus expansion for fertilization.

13. Supplemental Figure 8. Control groups without the hydrogel microcapsules need to be included in the experiments.

14. Discussion is redundant. It could be shortened by removing the parts already described in the results section.

15. Materials and methods. Lines, 850-852. The authors should measure luminescence signals, not fluorescence ones.

Reviewer #3:

Remarks to the Author:

General Comments:

The authors employed microencapsulation of mouse preantral follicles followed by vitrification in presence of low concentrations of cryoprotectants and laser- and electromagnetic-field assisted controlled ultra-rapid rewarming of capsules with nanoparticles to support and improve fertility preservation and survival of follicles. The paper presents convincing evidence that oocytes grown subsequently to antral stages and in vitro matured to metaphase II are of high quality and developmental potential and can develop to healthy pups after in vitro fertilization and transfer of embryos to foster mothers and that healthy offspring can be obtained from these after mating in vivo.

Overall, the study in the mouse model provides interesting and convincing data on the feasibility to use ultra-rapid nano-warming techniques in vitrification with low cryoprotectant to improve survival and vitality of preantral follicles in fertility preservation that might also open new approaches for restoring fertility in human.

Unfortunately, the presentation of data is not concise and it is difficult for readers to retrieve the main message about experimental set-up and rationale of using the mouse model. Furthermore, background of up-to-date knowledge on human fertility preservation and use of terms defining different stages of folliculogenesis are not precise or misleading and need to be revised, including citing more recent recommendations and up-to-date reports and reviews. Therefore, I cannot recommend accepting for publication in the present form but only after major revision.

For instance, recent advances in cryopreservation have addressed the potential harm to biological samples by ice crystal formation and multiple new approaches to increase efficiency and safety of cryopreservation (e.g. Chang and Zhao, *Adv Sci* 2021). These include use of different types of ice-inhibiting substances and membrane permeable or impermeable polymers, bioinspired cryoprotecting agents, hydrogels, nanoparticle delivery of cryoprotectants or nanocoating, e.g. with bioinspired materials and physical methods like magnetic, electro-magnetic field or laser thawing. The study has focused on preventing ice-crystal formation and damage during the re-warming procedure in presence of only low concentrations of cryoprotecting agents and is of relevance and interest.

Provided a revision is carefully conducted I highly recommend that the paper should be once more submitted and re-considered for publication in *Nature Communications* as it presents an innovative, novel concept for potentially successful novel methods that might be of interest in preservation of human and animal fertility. I would be happy to see it again for review as it has been carefully conducted using up-to-date novel methods to improve fertility preservation to scientists and clinicians.

Specific Comments:

Title should be shorter and more concise, e.g.:

Efficient cryopreservation in presence of low cryoprotectants of hydrogel micro-encapsuled mouse preantral follicles by novel rapid nano-warming techniques.

Abstract:

From the beginning it should be made clear that the major source of oocytes in mammalian ovary is in the primordial follicle pool that becomes depleted during aging or after gonadotoxic exposures. Such follicles are activated *in vivo* in a complex process and may also become activated inadvertently by stressors such as by freezing/ warming/ mechanochemical stressors. Homogenous populations of preantral follicles that possess a GV oocyte surrounded by one or few layers of granulosa cells can be conveniently obtained from hormonally stimulated prepubertal mice, or by young pubertal females and are therefore often used as a model for optimizing cryopreservation, *in vitro* culture conditions and fertility preservation in fertility research (Heiligentag et al., *Reprod Fertl Dev.* 2017; Xu and Zelinski, *Biol Reprod* 2021). Preantral follicles are also common in ovarian cortex of adult large mammals and used for preservation but not routinely used in human treatment.

Line 29: It is not mainly the cryopreservation of preantral but rather primordial follicles- preferably in ovarian tissue cryopreservation that is of relevance for fertility preservation in humans (e.g. Szymanska et al., *Mol Hum Reprod* 2020, Dolmans et al., *J Clin Med.* 2021; ESHRE Guideline: female fertility preservation Anderson et al., *U´Hum Reprod Update* 2020). Better state: for cryopreservation..is one promising..of fertility in mammals.

Success using follicle culture is difficult in human due to the long period of folliculogenesis and the

dramatic increase in size of maturing follicles not only in human but also other larger species like human, bovine, porcine or macaque. As stated by Gandoli et al., (Animal Reprod. 2019) rather "the use of ovarian fragments is the method for fertility preservation in human". Also, multiple step procedures are currently used in more experimental approaches which mainly involve primordial follicle activation and growth in slices, isolation of cumulus-oocyte complexes from large antral stages and in vitro maturation rather than preservation of pre-antral follicles (Telfer and Andersen, Fertil Steril 2021). Therefore, authors should from the beginning state that their aims of the present study is to improve cryopreservation techniques and avoid exposures to high cryoprotectants and mechanical and oxidative stress. Apparently, their interest is mainly in using the mouse model of preantral follicle culture to improve methods of cryopreservation by vitrification and advance encapsulation supporting good survival with a novel strategy to support follicle growth and oocyte maturation, a method that might have a potential to be applied also in human fertility preservation in the future.

Line 34: Better to state order of protocol: microencapsulation and vitrification followed by nonwarming using ...to achieve extremely rapid re-warming to prevent ice-crystal formation.

Line 35: It is also common for conventional vitrification of murine or larger species PAFs to achieve close to or over 90% survival (e.g. Trapphoff et al., Hum Reprod. 2010) state at which time after thawing the assays were performed and what percentage of recovered follicles produces MII oocytes after culture. In ovarian tissue cryopreservation follicles survived up to 98% according to other recent reports (El Cury-Silva et al., Cryobiology 2021). It is more important to state how many of the follicles yielded mature oocytes, pups and offspring after IVF, and compare this to controls using conventional vitrification methods.

Line 36: How many of these developed from the surviving follicles?

Introduction:

Line 51: Use more recent citations on increase in cancer risks such as: reports from the UK: Cancer Research UK Breast Cancer Incidence Statistics. [(accessed on 24 April 2019)]; Available online: <https://www.cancerresearchuk.org/health-professional/cancer-statistics/s>.

Line 54: Again, cite more recent literature (e.g. Szymanska et al., Mol Hum Reprod, 2020; Oktay et al., Fertil Steril 2022; Cacciottola et al., Best Pract Res Clin Obstet Gynaecol. 2021)

Line 58: Missing relevant recent citations! (ESHRE Guidelines fertility preservation, Anderson et al., Hum Reprod Open. 2020, Dolmans et al., J Clin Med. 2021 Etc.)

Line 71: This is simply not true! The cortex of ovary is particularly rich in primordial follicles and might be preserved in slices which can be transferred or in vitro cultured to obtain larger follicles from which mature IVM oocytes can be obtained (see for instance: discussion by Vo and Kuvamura, J Clin Med 2021).

Line 72: It is not preantral but primordial follicles that are the largest population in the ovary. Primordial follicles need to become activated to develop into pre-antral and antral follicles. Most of preantral follicles that become activated in vivo undergo demise and can therefore be rescued for fertility preservation.

Line 74: During primordial follicle ..

The authors should provide much better insights into procedures that can be used in fertility preservation, in particular in larger mammals. While it is possible to obtain large numbers of pre-antral follicles in prepubertal animals like the mouse after hormonal stimulation, this is certainly not possible in human. Such follicles can be generated from ovarian slices and subsequently cultured to larger stages from which oocyte cumulus-complexes can be derived and oocytes matured in vitro to metaphase II. However, the cryopreservation of preantral follicles would only be performed after activation of primordial follicles in slices of ovarian tissue, or is used to rescue in vivo activated PAFs

that may undergo atresia and is not common practice in human fertility preservation protocols.

Line 81 to 84: Again, authors do not correctly use the term preantral and primordial follicles. The latter are the most abundant whereas the formers are limited in numbers and present a larger sub-population of follicles that are not arrested but activated, can potentially mature but usually mostly become atretic. For animal fertility preservation activation to form preantral follicles can be used to isolate preantral stages however in pre-pubertal girls follicles may be isolated after activation in culture of slices.

Line 122: Add reference: Chang and Zhao, Adv Sci (Weinh) 2021

Line 129: oocytes and granulosa cells

Line 139: preovulatory antral follicles

Results:

It is difficult for readers to follow the procedures that are used from start to completion. Thus, Fig 1 and text does not contain information on encapsulation prior to vitrification and vitrification procedures in experimental pAFs and controls.

It appears that an open straw method has been used but it is difficult to retrieve the information whether EG, DMSO or other agents have been used for follicle vitrification in the control and sample. One has to search methods and read subsequent paragraphs to find the details of the experimental procedures before and during vitrification. Although the paper is focused on the re-warming steps, the information on how vitrified PAFs are obtained needs to be given first. It is only evident from Fig. 7 that PAF-alginate encapsulation has been performed before follicles were vitrified. Therefore the paper would greatly benefit from a flow chart that might be added as supplement or better at beginning of the result section providing information on culture medium, alginate concentration and cryoprotectants before and during vitrification and on all washing solutions and rewarming steps used during the re-warming process.

Line 151: Figure 1: Explain abbreviation in figure legend in this as in all other figures: e.g. NPs: nanoparticles, GO etc

Line 161: This information should be given in abstract to make clear that the paper compared the survival and developmental capacity of alginate encapsulated PAFs that were re-warmed by different methods to prevent ice-crystal formation under low CPA concentrations.

Line 166: may? Better: to explore improved vitality and developmental competence of cryopreservation with low cryoprotectant.

Line 252: (0,green, 1M light green, 1.5M yellow, 2M blue bars)..different CPAs (vertical axis)..

Line 254: Unclear legend. Please, refer to color in groups.

Line 323: Rate of MII for all cultured follicles or only those maturing to antral stage?? Please, add numbers to each panel.

Line 334: .. appeared in % of PAFs.. % of PAFs (n=...respectively)

Line 351: cultured pre-antral follicles?? Of similar size??

Line 355: Omit this sentence. In the study it is not the MII oocytes that are vitrified warmed and therefore formation of spindles is just reflecting normal maturation of oocytes after culture of follicles.

Line 370: Better: Functional mitochondria are essential for normal oogenesis and embryogenesis.

Line 372: add more recent citations such as De Cota et al., *Reprod Fert Dev*2020; Iwata , *Reprod. Med Biol* 2021

Line 382: Add information on which oocytes have been analyzed: The GV oocytes after re-warming or rather MII oocytes after culture? This is vital as MMP may transiently decrease and then recover; see and add citation: Demand et al., 2012

Line 393 and following: always state stage of oocytes, GV or MII, from re-warming or after culture and growth?

Line 415, Fig. 5: Alter blastosphere to blastocyst

Figure 6c: There appear to be always twice as many male compared to female pups after 2-cell transfer- please explain!

Line 423: needs to be..

Discussion:

Line 499: This is a most relevant improvement that should be pointed out in the abstract

Line 502: only with low CPA- traditional warming with high CPA and washing steps has much higher survival rates e.g.

Line 527: This is a comparative short time for toxicity test and remaining NP should be analyzed in future studies.

Line 538: It may depend on size of follicles in different species whether microencapsulation is favorable or restricts follicle and oocyte expansion and growth.

Line 558: this may be true for the mouse model- however, follicle growth in larger mammals may be not so ideal when there is more mechanical stress and possible problems with oxygen diffusion.

Line 572: the MII oocytes

Line 591: Omit sentence. The successful..Better: Adaptation of the novel protocols for cryopreservation for human ovarian material and PAFs may help improve...

M&M:

Line 616: of ?? week old ..

Line 706: Which concentration of CPA was used?

Line 726: removed by??

Line 898: 6 embryos each??

Consider to add references, e.g.

Cancer Research UK Breast Cancer Incidence Statistics. [(accessed on 24 April 2019)]; Available online: <https://www.cancerresearchuk.org/health-professional/cancer-statistics/s>.

Cacciottola L, Donnez J, Dolmans MM. *Best Pract Res Clin Obstet Gynaecol*. 2021 Nov 15;S1521-6934(21)00165-6. doi: 10.1016/j.bpobgyn.2021.09.010. Online ahead of print. PMID: 34887172

El Cury-Silva T, Nunes MEG, Casalechi M, Comim FV, Rodrigues JK, Reis FM. *Cryobiology*. 2021 Dec;103:7-14. doi: 10.1016/j.cryobiol.2021.08.001. Epub 2021 Aug 8. PMID: 34370991
Best Pract Res Clin Obstet Gynaecol. 2021

ESHRE Guideline Group on Female Fertility Preservation, Anderson RA, Amant F, Braat D, D'Angelo A, Chuva de Sousa Lopes SM, Demeestere I, Dwek S, Frith L, Lambertini M, Maslin C, Moura-Ramos M, Nogueira D, Rodriguez-Wallberg K, Vermeulen N. *Hum Reprod Open*. 2020 Nov 14;2020(4):hoaa052. doi: 10.1093/hropen/hoaa052. eCollection 2020. PMID: 33225079

Del Castillo LM, Buigues A, Rossi V, Soriano MJ, Martinez J, De Felici M, Lamsira HK, Di Rella F, Klinger FG, Pellicer A, Herraiz S. *Hum Reprod*. 2021 Aug 18;36(9):2514-2528. doi: 10.1093/humrep/deab165. PMID: 34333622

Telfer EE, Andersen CY. *Fertil Steril*. 2021 May;115(5):1116-1125. doi: 10.1016/j.fertnstert.2021.03.004. Epub 2021 Apr 3. PMID: 33823993

Rodriguez-Wallberg KA, Marklund A, Lundberg F, Wikander I, Milenkovic M, Anastacio A, et al. . *Acta Obstet Gynecol Scand* (2019) 98(5):604–15. doi: 10.1111/aogs.13559 - DOI - PubMed

Laronda MM, Rutz AL, Xiao S, Whelan KA, Duncan FE, Roth EW, Woodruff TK, Shah RN. *Nat Commun*. 2017;8:15261. doi: 10.1038/ncomms15261.

Ting AY, Yeoman RR, Campos JR, Lawson MS, Mullen SF, Fahy GM, Zelinski MB. *Hum Reprod*. 2013;28(5):1267–1279. doi: 10.1093/humrep/det032.

Moravek MB, Appiah LC, Anazodo A, Burns KC, Gomez-Lobo V, Hoefgen HR, Jaworek Frias O, Laronda MM, Levine J, Meacham LR, Pavone ME, Quinn GP, Rowell EE, Strine AC, Woodruff TK, Nahata L. . *J Adolesc Health*. 2019;64(5):563–573. doi: 10.1016/j.jadohealth.2018.10.297.

Szymanska et al., *Mol Hum Reprod* 2020 Aug 1;26(8):553-566.
doi: 10.1093/molehr/gaaa043.

Kim et al. *Theriogenology* 2020 Mar 1;144:33-40.
doi: 10.1016/j.theriogenology.2019.12.009.

Sugishita et al *Hum Reprod* 2021 Dec 20;deab274.
doi: 10.1093/humrep/deab274. Online ahead of print.

Gupta et al. *Theriogenology* . 2022 Jan 15;178:1-7.
doi: 10.1016/j.theriogenology.2021.10.024. Epub 2021 Oct 25.

Wu et al. *Int J Biol Macromol* 2021 Dec 1;192:1276-1291.
doi: 10.1016/j.ijbiomac.2021.09.211. Epub 2021 Oct 9.

Chang and Zhao *Adv Sci (Weinh)* 2021 Feb 1;8(6):2002425.

doi: 10.1002/adv.202002425. eCollection 2021 Mar.

Delattre et al., . 2020 Nov 1;35(11):2524-2536.
doi: 10.1093/humrep/deaa193.

Response to Reviewers' comments:

We are very grateful to the referees for their insightful and thoughtful comments. In this revision, we have carefully addressed all the comments, which significantly improves the clarity and quality of the manuscript. The point-by-point responses to all the comments are listed below. All changes for addressing the comments are also highlighted in yellow in the revised manuscript

Reviewers' comments:

Reviewer #1:

Comments to the Author

The manuscript shows very interesting and successful results of mouse preantral follicles (PAFs) vitrification associating microencapsulation in alginate and nano-warming using magnetic and light induced hyperthermia by mixing iron oxide and graphene NPs to the vitrification solution. With this approach, the authors present a protocol that allows high survival of vitrified PAFs, which are able to develop in vitro to antral follicles and produce viable oocytes that can be in vitro fertilized and produce live young after embryo transplantation to surrogate mothers. The study is elegant and very complete. Methodology and results are sound, and figures are informative and self-explanatory. The study is well based on pertinent literature.

The study brings a novel methodology that may be of significance to the field of animal and human fertility preservation.

Only a few points must be considered to make the manuscript clearer.

[Response]: We thank the reviewer for these encouraging comments. We have revised and modified the main text to address your concerns, and detailed corrections are listed below, point by point. Hopefully, you will find it justified.

Major points:

1. Please, change “*in situ*” to ‘*in vitro*’ throughout the text (there are 9 places where the term “*in situ*” is used wrongly, including the title).

[A1]: We thank the reviewer for pointing this out. As per the advice, we have fixed “*in situ*” with “*in vitro*” in the manuscript.

2. Lines 211-214 - Please, include the duration (in time units) of ice crystals in each of the 4 treatment groups in the text.

[A2]: We thank the reviewer’s insightful suggestion. As per advice, the duration (in time units) of ice crystals in each of the 4 treatment groups has been added to the manuscript as follows (Lines 209-213, pages 11-12 in this revision):

Compared with those of PS #1, the ice crystal residence time was shorter in PS #2 with MIH and PS #3 with LIH (4.2 s vs. 2.96 s and 2.92 s, Supplementary Movies 2, 3 and

4). As expected, PS #4 with MIH and LIH showed the shortest ice crystal duration (2.40 s, Supplementary Movie 5).

3. Lines 266-267 - Where are the statistics of this result? The figure doesn't show statistical differences among CPA concentration groups.

[A3]: Thank you for pointing out this question.

i. The statistics of this result have been added to the **Fig. 3** legends.

ii. There were no statistical differences in the survival rate of PAF among CPA concentration groups. Just because the value of PAF survival rate was slightly higher when using 1.5 M CPA compared with 1 M or 2 M CPA, we recommended 1.5 M CPA. As per the advice, **Fig. 3b** was split into **Fig. 3b** and **c** to more clearly show the statistical analysis of the results.

The sentences were revised in the manuscript as follows (Lines 288-291, page 16 in this revision):

There were no statistical differences in the survival rate of PAFs among the different CPA concentration groups. However, the value of PAF survival rate with 1.5 M CPA was slightly higher than that with 1 M and 2 M CPA (Fig. 3b).

4. Lines 290-300 - There are no statistical differences to support those affirmations, at least not shown on the Figure. If there is any statistical significance among the different current /light intensities, please include on the manuscript. I don't see any problem in the choice being random in this case, but it must be clear in the text.

[A4]: Apologize for the negligence, and thank you for your suggestion. There was no statistical significance among the different current /light intensities. Just because the value of PAF survival rate was slightly higher under 15 A current intensity and 3 W/cm² light intensity than other groups, we recommended this condition.

The statistics of this result have been added to **Supplementary Fig. 5** legends.

Supplementary Fig. 5a was split into **Fig. 5b** and **c**, **Supplementary Fig. 5b** was split into **Fig. 5c** and **d** to more clearly show the statistical analysis of the results.

As per advice, the sentences were revised in the manuscript as follows (Lines 313-323, page 17 in this revision):

In addition, PAF survival rates changed with different current intensities (10 A, 15 A, 20 A) and different combinations of Fe₃O₄ and GO NPs (0.2% and 0.03%, 0.3% and 0.03%, 0.4% and 0.03%, w/v). Results showed that 0.3% Fe₃O₄ was the most effective (Supplementary Fig. 5a). Besides, GO concentration and laser intensity also affected the survival rate of PAFs. When keeping 0.3% Fe₃O₄ unchanged, the best result was obtained with 0.03% GO, compared with 0.02% and 0.04% GO (Supplementary Fig. 5c). Although there was no statistical significance among the different current and laser intensities, we found that the value of PAF survival rate was slightly higher at 15 A and 3 W/cm² than other groups (Supplementary Fig. 5b and d).

5. Lines 401-403 - I don't see this difference for MS in the figure (for instance, a similar

difference in the bars size is observed in **Fig. 8i** and you say they were similar in lines 409-410). For ROS the graphic's bar is a little smaller, but there is no statistical difference indicated. The only significant difference on mitochondrial function among Fresh, W/O VIF and W/ MIH+LIH is on ATP level (**Fig. 8c**).

Authors should stick to statistical differences and be consistent throughout the text.

[A5]: Apologize for the inconvenience and thank you for your suggestion. There was no significant difference in MS, ROS, and calcium among Fresh, W/O VIF, and W/ MIH+LIH. The statistics of this result have been added to **Supplementary Fig. 10** legends.

As per the advice, the sentence was revised in the manuscript as follows (Lines 432-435, page 23 in this revision):

There were no statistically significant differences in MS and ROS levels in oocytes among the Fresh, W/O VIF and W/MIH+LIH groups (Supplementary Fig. 10g and h).

6. Pg 47-49 – I think the Methods for detection of MMP, ATP, MS, ROS and Ca⁺ levels could be described together, since they all refer to mitochondrial function and because some sentences are repeated several times on those topics (for example: “Then, oocytes were placed in 96-well black assay plates (6 oocytes/well) (3922, Corning) after washing three times with PBS” , “the relative fluorescence of each oocyte was calculated”, "images of oocytes were obtained by confocal laser scanning microscopy”).

[A6]: We appreciate the reviewer’s insightful suggestion. As per the advice, the methods for detection of MMP, ATP, MS, ROS, and Ca⁺ levels were described together in the manuscript as follows (Lines 855-873 pages 46-47 in this revision):

Evaluation of Oocyte MMP, ATP, MS, ROS and Calcium Ions Levels

To test the mitochondrial function of oocytes, oocytes were divided into five major groups (each group contained three groups: Fresh, W/O VIT and W/ MIH+LIH). In order to assess the levels of MMP, ATP, MS, ROS and calcium ions in oocytes, the oocytes were incubated in JC-1 staining solution (KGA601, KeyGen, China), CellTiter-Glo Reagent (Luminescent cell viability assay kit, Promega), MitoSOX Red Mitochondrial Superoxide Indicator (M36008, Invitrogen), DCFH-DA (D6470, Solarbio, China) staining solution, and Fluo-4 AM (KGAF024, Keygen, China) for 20-30 min respectively at 37°C in a 5% CO₂ incubator. Then, oocytes were placed in 96-well black assay plates (6-8 oocytes/well) (3922, Corning) after washing three times with PBS. Subsequently, the fluorescence/luminescence intensity was measured by a multimode reader (SpectraMax iD5, Molecular Devices). And the fluorescence/luminescence intensity was divided by the number of oocytes to calculate the relative fluorescence value of each oocyte. Among them, the ratio of red/green fluorescence intensity of each oocyte was used as an indicator to measure MMP. Finally, fluorescence images of oocytes were taken by confocal laser scanning microscopy (LSM 710, Zeiss, Germany).

Minor points:

We agree and thank the reviewer's insightful suggestion. We have revised and modified the main text to address your concerns, and detailed corrections are listed below, point by point. We hope that it is now clearer.

1. Line 56 – Change “fertility in women preservation” to ‘fertility preservation in women’

[A1]: Apologize for the negligence, and thank you for your suggestion. As per the advice, the “fertility in women preservation” has been changed to “fertility preservation in women” in the manuscript (Lines 51-52, page 3 in this revision).

2. Line 67 – Change “transplanted ovarian tissue” to ‘ovarian tissue to be transplanted in the future’

[A2]: Apologize for the negligence, and thank you for your suggestion. As per the advice, the “transplanted ovarian tissue” has been changed to “ovarian tissue to be transplanted in the future” in the manuscript (Lines 61-63, page 3 in this revision).

3. Line 102-103 – The problems pointed out for IRIs apply to NPs as well. Please, delete “almost all IRIs are exogenous, non-natural, non-Food and Drug Administration (FDA) approved additives. Moreover,”

[A3]: Thank you for your suggestion. As per the advice, “almost all IRIs are exogenous, non-natural, non-Food and Drug Administration (FDA) approved additives. Moreover,” has been removed from the manuscript.

4. Line 129 – Change “granule/membrane cells” to ‘granulosa cells’

[A4]: Thank you for your suggestion. As per the advice, the “granule/membrane cells” has been changed to “granulosa cells” in the manuscript (Lines 114-117, page 6 in this revision).

5. Line 205 – Please change “transparent” to translucent

[A5]: Thank you for your suggestion. As per the advice, the “transparent” has been changed to “translucent” in the manuscript (Lines 203-205, page 11 in this revision) and **Fig. 2b(ii)**.

6. Line 217 - Inhibit or minimize? I think I can still see some ice crystal formation in the pictures and the video.

[A6]: Apologize for the negligence, and thank you for your suggestion. As per the advice, the sentence was revised in the manuscript as follows (Lines 216-218, page 12 in this revision):

Nano-warming combining MIH and LIH may minimize devitrification/recrystallization and improve the quality of PAFs after low-pCPA vitrification and warming.

7. Line 307 - What do you mean by “membrane cells”? Do you mean basement membrane? There are no cells on the basement membrane. Please, clarify.

[A7]: Apologize for the inconvenience and thank you for your comment. As per the advice, the sentence was revised in the manuscript as follows (Lines 328-330, page 17-18 in this revision):

The normal vitrified PAFs had a central round oocyte, intact granulosa cell layer and basement membrane that were the same as those of fresh PAFs.

8. Lines 315-316 - Not for Fgf8 - there is a statistical difference appointed on the figure.
[A8]: Apologize for the negligence, and thank you for your suggestion. As per the advice, the sentence was revised in the manuscript as follows (Lines 340-341, page 18 in this revision):

There was no significant difference on the expression of the Gdf9, Fgf8 and Bmp15 between W/ MIH+LIH and Fresh .

9. Lines 322 and 348 – Change “probability” to ‘percentage’

[A9]: Thank you for your suggestion. As per the advice, the word “probability” has been changed to “percentage” in the manuscript (Lines 348, page 19 in this revision).

10. Line 325/Figure 4 - The nomenclature of the groups is a little confuse. Please, explain on the figure caption what exactly are W/O VTF and W/ LIH+MIH as well.

[A10]: Apologize for the inconvenience and thank you for your suggestion. As per the advice, the explanation of W/O VTF and W/ LIH+MIH has been added to the **Fig. 4** caption in the manuscript as follows (Lines 351-354, page 20 in this revision).

W/O VTF: PAFs were only encapsulated in hydrogel and did not undergo vitrification; W/ MIH+LIH: PAFs encapsulated in hydrogel were warmed with MIH and LIH after vitrification.

11. Line 339 – Change “proliferation” to ‘growth’

[A11]: Thank you for your suggestion. As per the advice, the word “proliferation” has been changed to “growth” in the manuscript (Lines 375, page 21 in this revision).

12. Line 340 – Change “of the PAFs” to ‘of the vitrified and warmed by MIH+LIH PAFs’

[A12]: Thank you for your suggestion. As per the advice, the “of the PAFs” has been changed to “of the vitrified and warmed by MIH+LIH PAFs” in the manuscript (Lines 376-378, page 21 in this revision).

13. Lines 355-359 – This statement is true but does not apply to this study since mature oocytes were not cooled/warmed. The spindle analysis is important, but this piece of text is not necessary. Please, delete.

[A13]: Thank you for your suggestion. As per the advice, the sentences “The spindle is easily disrupted during cooling and warming due to changes in temperature and ice

crystal formation. The breakage of the spindle usually causes chromosomal disorganization, which adversely affects the euploidy of the embryo after fertilization.” has been removed from the manuscript.

14. Figure 6 - What do you mean by “Birth sequence” ? Please, clarify.

[A14]: We thank the reviewer for pointing this out and apologize for our error. It is inaccurate to use “Birth sequence” here. The “Birth sequence” has been modified as “Surrogate Mouse Number” in **Fig. 7c** and **e**.

15. Line 435 - Change “probability” to ‘rate’

[A15]: Thank you for your suggestion. As per the advice, the “probability” has been changed to “rate” in the manuscript (Lines 495-496, page 28 in this revision).

16. Lines 446-447 – Change “the reproductive performance” to ‘development’

[A16]: Thank you for your suggestion. As per the advice, the “the reproductive performance” has been changed to “development” in the manuscript (Line 514, page 29 in this revision).

17. Line 450 – I think you meant second generation (F2) instead of third generation. Please, correct.

[A17]: Apologize for the inconvenience, and thank you for pointing this out. As per the advice, the sentence has been revised in the manuscript (Lines 516-518, page 29 in this revision).

18. Line 469 – Change “atrophied” to ‘shrunk’

[A18]: Thank for your suggestion. As per the advice, the “atrophied” has been changed to “shrunk” in the manuscript (Lines 534-536, page 31 in this revision).

19. Line 574 – Include ‘meiotic’ before “spindles”

[A19]: Thank you for your suggestion. The relevant sentences have been removed from the manuscript to streamline the discussion.

20. Line 586 - Include ‘in vitro’ before “fertilized”

[A20]: Thank you for your suggestion. As per the advice, the “*in vitro*” has been added before “fertilized” to the manuscript (Lines 634-635, page 35 in this revision).

21. Line 620 - Were the PAFs dissected individually? Or was the tissue macerated and PAF harvested afterwards? Please, clarify in the text.

[A21]: Thank you for your suggestion. The PAFs were dissected individually by two 1 mL syringe needles. Prior to this, the ovarian tissue had not been treated with any lysis-related enzymatic solutions.

As per the advice, the sentences have been revised to the manuscript (Lines 663-664, page 37 in this revision).

22. Line 621 - It would be better to mention the Gauge of the used needle.

[A22]: Thank you for your suggestion. The Gauge of the used needle was 0.45 mm (0.45×16 RWLB). As per the advice, the sentence has been added to the manuscript as follows (Lines 667-669, page 38 in this revision).

Then, PAFs were immediately separated from ovaries using two 1 mL syringe needles (0.45×16 RWLB) and moved to the prepared separation medium for microencapsulation.

23. Line 681 - PAFs' TEM (showed in Supplementary Figure 3) methodology was not described in the text.

[A23]: Apologize for the negligence, and thank you for your suggestion. As per the advice, PAFs' TEM methodology has been added to the manuscript (Lines 730-746, page 41 in this revision).

24. Line 759 - Supplementary Table 1 only brings the primer sequences of 3 genes, and the results shown are also for 3 genes only. Please, correct the text accordingly.

[A24]: Apologize for the negligence, and thank you for your suggestion. As per the advice, the “four” has been modified as “three” in the Supplementary Information (Lines 209-210, pages 15-16 in this revision).

25. Line 783 – Please, change “vertical” to ‘cross’

[A25]: Thank you for your suggestion. As per the advice, the “vertical” has been changed to “cross” in the manuscript (Line 815, page 42 in this revision).

Supplementary Material:

In general, supplementary material is helpful and well connected to the manuscript. Supplementary Figure 1 and Supplementary Movie 1 are very helpful for understanding the alginate microcapsules formation and structure.

[Response]: We are grateful for these encouraging comments.

Minor points:

1. Supplementary Movies 2-5 could be in slow motion to allow a better visualization of the warming process.

[A1]: Thank you for your insightful suggestion. As per the advice, the playback speed of **Supplementary Movies 2-5** has been slowed down in the Supplementary Information.

2. Supplementary Figure 6 – The figure says the follicle is fresh, while the figure caption says “PAF-alginate hydrogel microcapsules dissolved in 75 mM sodium citrate” and the text (lines 336-337) says “when the PAFs were not encapsulated by hydrogels”. Was this follicle encapsulated while fresh and then the microcapsule dissolved before placing in IVC, or not encapsulated at all? Please, be more specific and coherent.

[A2]: Apologize for the negligence, and thank you for your suggestion. This was fresh PAF without (W/O) encapsulation. The sentence has been modified to the manuscript

(Lines 371-374, pages 20-21 in this revision) and **Supplementary Fig. 7** caption.

However, the 3D shape of the follicles could not be retained, and the granular cells diffused to the surroundings when fresh PAFs without (W/O) encapsulation were directly cultured in a 2D environment (Supplementary Fig. 7).

We would like to thank the reviewer again for taking the time to review our manuscript.

Reviewer #2 (Remarks to the Author):

Cryopreservation of PAFs is a promising method for fertility preservation. Although vitrification has potentials to become a superior approach, it still needs further improvement. The authors focused on the thawing step of cryopreserved PAFs by assessing the beneficial effects of the alginate hydrogel microcapsules as well as the nanowarming using NPs. Whilst interesting, there are below concerns need to be addressed.

[Response]: We are grateful for the insightful and thoughtful comments.

Major concerns

1. It is overstating to mention about clinical application. Due to large size of preovulatory antral follicle and long duration from PAF to preovulatory antral follicle for development in human, the concept of *in vitro* culture of PAFs shown in this manuscript is difficult to apply. The hydrogel-microcapsuled PAFs may be able to graft into patient ovaries, the hydrogel microcapsules could prevent vascular reconstruction to the follicles.

[A1]: Apologize for the inconvenience and thank you for your insightful suggestion. As per advice, the contents related to the clinical application have been removed from the manuscript.

2. Many experiments are lacking the control groups without hydrogel microcapsules. Furthermore, the number of samples in most of experiments is unclear. Although some experiments indicate the sample numbers, they are too small (n=3) to lead the authors' conclusion.

[A2]: Thank you for your valuable suggestion.

i. The fresh PAFs without (W/O) encapsulation could develop into antral follicles after 2D culture, as shown in **Supplementary Fig.7**. Unfortunately, the granulosa cells around the PAF were heavily diffused, and oocytes could not be collected. Oocytes could be successfully collected by culturing the hydrogel-encapsulated PAF *in vitro*. The hydrogel-encapsulated PAFs were selected as the control group in the subsequent characterization of follicle function. So, there was no group without hydrogel microcapsules in the subsequent experiments.

ii. Apologize for the negligence, and thank you for your suggestion. Here n=3 meant that the experiment was replicated three times. Statistical analysis of the results and sample numbers were added to the legends of each figure for a clearer presentation of the experimental sample numbers.

3. Evaluation of epigenetic changes in oocytes/embryos recovered after this approach is missing.

[A3]: Thank you for your suggestion. As per the advice, evaluation of histone modifications changes in oocytes/embryos recovered after this approach has been added to the manuscript as follows (Lines 481-492, page 27 in this revision) and **Fig. 6 and Supplementary Fig. 11**.

Epigenetic Analysis of Embryos at Different Stages

As one of the most important epigenetic indicators, histone modifications are of great significance in controlling gene expression and regulating early embryonic development⁵⁶. H3K9me3, H3K4me3 and H3K27ac are typical histone modifications⁵⁷⁻⁵⁹, so we evaluated them by fluorescent staining (Fig. 6 and Supplementary Fig. 11). The results showed that H3K9me3, H3K4me3 and H3K27ac were similar in W/MIH+LIH and W/O VTF (Fig. 6 and Supplementary Fig. 11). Among them, H3K9me3 on the paternal genome was erased at the time of fertilization and could be gradually restored afterward. Therefore, H3K9me3 from the paternal parent was not seen during the two-pronuclei stage and could only be observed after the two-cell embryonic stage.

56. Weaver, J. R., Susiarjo, M. & Bartolomei, M. S. Imprinting and epigenetic changes in the early embryo. *Mamm Genome* 20, 532–543 (2009).

57. Huang, X. et al. Stable H3K4me3 is associated with transcription initiation during early embryo development. *Bioinformatics* 35, 3931–3936 (2019).

58. Wang, C. et al. Reprogramming of H3K9me3-dependent heterochromatin during mammalian embryo development. *Nature cell biology* 20, 620–631 (2018).

59. Marinho, L. S. R., Rissi, V. B., Lindquist, A. G., Seneda, M. M. & Bordignon, V. Acetylation and methylation profiles of H3K27 in porcine embryos cultured in vitro. *Zygote* 25, 575–582 (2017).

4. No data is shown to prove whether NPs are completely separated from PAFs.

[A4]: Thank you for your suggestion. As per the advice, energy dispersive X-ray spectroscopy (EDS) of the hydrogel microspheres and uptake detection of NPs were added in the manuscript as follows (Lines 245-258, pages 13-14 in this revision) and **Supplementary Fig.4**.

In addition, energy dispersive X-ray spectroscopy (EDS) of the hydrogel microspheres was performed to examine whether the NPs could be completely separated from PAFs. The results showed that carbon, oxygen, calcium and iron elements could be observed on the surface of the hydrogel microspheres after incubation with NPs, but no iron element can be detected inside the hydrogel. After washing, the iron element could no longer be observed on the surface and inside of the hydrogel microspheres (Supplementary Fig. 4a). This indicated that some Fe₃O₄ NPs adhered to the surface of the hydrogel microspheres, but could not enter the interior of the hydrogel after incubation with NPs. Moreover, the Fe₃O₄ NPs could be washed away completely. In addition, the TEM results of PAFs also showed that no Fe₃O₄ NPs were taken in by PAFs (Supplementary Fig. 4b and c). From all the above results, it could be seen that the PAFs and NPs were separated by the hydrogel microspheres.

Individual concern

5. The reviewer does not agree that PAFs can be obtained from the ovaries of old women. Indeed, the number of PAFs in old women is very limited and difficult to isolate due to excess fibrosis in ovaries.

[A5]: We agree and thank the reviewer for pointing this out. As per the advice, “Additionally, PAFs are present in the reproductive glands of individuals of all ages and can be obtained from the ovaries of young or old women, which is impossible to achieve in oocyte cryopreservation.” has been removed from the manuscript.

6. Figure 2B. The explanation of “****” in the graph is lacking.

[A6]: Apologize for the negligence, and thank you for your suggestion. As per the advice, the explanation of “****” has been added in **Fig. 2** legends.

7. Figure 2. High magnification images of PAFs in the hydrogel microcapsules need to be included.

[A7]: We appreciate the reviewer’s insightful suggestion. **Fig. 2** mainly introduced the thermal effect of Fe₃O₄ NPs and GO NPs. This part of the experiment did not use hydrogel-encapsulated PAFs, only CPAs with NPs. So there was no need to add high magnification images of PAFs in the hydrogel microcapsules.

8. Supplemental Movie 1. In this movie, PAF locates in the peripheral position of the hydrogel microcapsules. It is unclear if such location of PAF affects the efficiency of nano-warming and following *in vitro* culture procedures. The movie shows some abnormal shape microcapsules (not perfect sphere) without PAFs. The data is not given whether this abnormal shape microcapsules interferes the outcome of experiments. Based on the movie and materials and methods, multiple PAFs could be encapsulated in one microcapsule and such PAFs might be difficult to develop after thawing and cultures. Together with considering the formation of abnormal shape microcapsules, the success rate of preparation of proper PAFs in the hydrogel microcapsules should be presented.

[A8]: Thank for your suggestion.

i. In this experiment, nano-warming was performed by loading NPs into CPA solution, and then heating the PAFs encapsulated in hydrogel microcapsules under an alternating electromagnetic field and a near-infrared laser. The hydrogel microcapsules with a diameter of about 400 μm were small and did not affect the heating effect. The microcapsules maintain the three-dimensional structure of PAFs *in vitro*, mainly through the 3D network structure of the hydrogel. The location of PAFs on the periphery of the hydrogel microcapsules did not affect the subsequent *in vitro* culture process. Therefore, the location of the PAF at the peripheral position of the hydrogel microcapsules did not affect the nano-warming and following *in vitro* culture procedures.

ii. Because PAF-encapsulated hydrogel microspheres were selected for subsequent freezing experiments in this study, these abnormally shaped microcapsules without PAF would not interfere with the outcome of experiments.

iii. After warming, only hydrogel microcapsules encapsulating one PAF were selected for subsequent culture experiments. Therefore, microcapsules encapsulating multiple PAFs did not affect subsequent PAF development.

iv. The number of PAFs that could be used in each experiment was not large (20-30), and the centrifugal microfluidic device could generate hundreds of hydrogel

microspheres at a time, so the rate of hydrogel microcapsules encapsulating one PAF was not high. The success rate of preparation of one PAF in the hydrogel microcapsules was about 20%.

9. Movies 2-5. These movies are not suitable for evaluating the ice-crystal formation.

[A9]: We appreciate the reviewer's concern about **Movies 2-5**. It mainly detected the duration of ice crystals during the warming to prove that nano-warming combined with MIH and LIH could shorten the residence time of PAFs in the dangerous temperature zone and reduce the damage of ice crystals to cells. These movies are not evaluating ice crystal formation. We have slowed down the playback speed of **Supplementary Movies 2-5** to make it clearer to observe the duration of the ice crystals.

10. Supplemental Figure 3. The reviewer could not detect the hydrogel microcapsules in pictures. (b) needs positive controls and another control group without the hydrogel microcapsules. (c) needs explanation of pictures.

[A10]: Thank you for your suggestion.

(a) As per advice, the hydrogel microcapsules encapsulated with PAF were added in **Supplementary Fig. 3**.

(b) The positive controls and another control group without the hydrogel microcapsules were added in **Supplementary Fig. 3**.

(c) The explanation of pictures has been added to the manuscript as follows (Lines 239-245, page 13 in this revision).

In addition, the biocompatibility of NPs was tested, as shown in Supplementary Fig. 3. The results showed that the survival rates of PAFs with (W/) or without (W/O) encapsulation in the medium with (W/) or without (W/O) NPs were basically similar after incubation for different times (0, 6, 12, 24, 36, 48, 60 h) at 37°C. This suggested that neither NPs nor hydrogels were toxic to PAFs.

11. Supplemental Figure 5. Similar to the Supplemental Figure 3, The reviewer could not detect the hydrogel microcapsules in pictures. Control groups without the hydrogel microcapsules need to be include in the experiments. There is no explanation why the authors measured the transcript levels of oocyte-derived factors on day 13 of cultures.

[A11]: Thank for your suggestion.

i. Fluorescence images of PAFS encapsulated in hydrogel microcapsules have been added to **Supplementary Fig. 6a**.

ii. **Supplementary Fig. 6a and b** mainly showed the fluorescence images of normal and abnormal PAFs under high magnification, so we did not add the fluorescence images of the group without the hydrogel microcapsules.

iii. On day 13, antral follicle development was stable. So, the three genes expression levels of the antral follicles on day 13 were measured.

12. Figure 4. (a) the explanation of arrow is lacking. The data shown in Supplemental Figure 6 need should be shown here. (c and d) The P₄ and E₂ levels are originate from

one follicle based on the materials and methods. If so, they are too high. Also, it seems to be difficult that the PAFs continue to produce P4 and E2 for 12 days without adding androgen substrate. The pictures of COCs obtained from cultured PAFs should be included to demonstrate the normality of cumulus expansion for fertilization.

[A12]: Apologize for the negligence and thank you for your suggestion.

i. As per advice, the explanation of the arrow was added in **Fig. 4 legends**.

ii. **Supplementary Fig. 7** showed the results of two-dimensional (2D) culture of PAFs without encapsulation. Since intact oocytes could not be collected after 2D culture of PAFs, PAFs encapsulated in hydrogel microspheres were used as a control group for subsequent experiments. So, there is no need to move **Supplementary Fig. 7** here.

iii. Apologize for the negligence. In the original figure, the secretion of E2 and P4 came from 10~30 follicles. For a clearer view of the secretion of the two hormones, the average secretion per follicle was analyzed in **Fig. 4c** and **Supplementary Fig. 8**. And the detailed detection process has been added in this manuscript (Lines 378-381, page 21; Lines 813-825, pages 44-45 in this revision).

iv. It has been demonstrated in many studies that the secretion of E2 and P4 could be detected in the culture medium of PAFs after 10~13 days without the addition of androgens *in vitro*¹⁻³. Previous researchers have confirmed that the secretion of E2 and P4 on the 12th day is not difficult, and it was indeed detected in our experiment.

1. Agarwal, P. et al. A biomimetic core-shell platform for miniaturized 3D cell and tissue engineering. *Particle & Particle Systems Characterization* 32, 809–816 (2015).

2. Jamalzaei, P., Valojerdi, M. R., Montazeri, L. & Baharvand, H. Effects of alginate concentration and ovarian cells on in vitro development of mouse preantral follicles: a factorial study. *International Journal of Fertility & Sterility* 13, 330 (2020).

3. Laronda, M. M. et al. A bioprosthetic ovary created using 3D printed microporous scaffolds restores ovarian function in sterilized mice. *Nat Commun* 8, 15261 (2017).

v. As per advice, the pictures of COCs obtained from cultured PAFs were added in **Fig. 4d**, and explanations of them were added in this manuscript as follows (Lines 386-387, page 21 in this revision).

The pictures of COCs obtained from cultured PAFs were shown in the Fig. 4d. It could be seen that COCs was normal.

13. Supplemental Figure 8. Control groups without the hydrogel microcapsules need to be included in the experiments.

[A13]: Thank you for your valuable suggestion. Since intact oocytes could not be collected after 2D culture of PAFs, PAFs encapsulated in hydrogel microspheres were used as a control group for subsequent experiments. **Supplementary Fig. 10** mainly showed some functional assays of oocytes. So, W/VTF and fresh oocytes were used as controls, and the groups without the hydrogel microcapsules were not included in the experiments.

14. Discussion is redundant. It could be shortened by removing the parts already described in the results section.

[A14]: Thank you for your valuable suggestion. As per advice, the parts already described in the results section have been removed from the discussion in this manuscript.

15. Materials and methods. Lines, 850-852. The authors should measure luminescence signals, not fluorescence ones.

[A15]: Apologize for the negligence, and thank you for your suggestion. As per advice, the sentences were revised in the manuscript as follows (Line 866-870, page 47 in this revision).

Subsequently, the fluorescence/luminescence intensity was measured by a multimode reader (SpectraMax iD5, Molecular Devices). And the fluorescence/luminescence intensity was divided by the number of oocytes to calculate the relative fluorescence value of each oocyte.

We would like to thank the reviewer again for taking the time to review our manuscript.

Reviewer #3 (Remarks to the Author):

General Comments:

The authors employed microencapsulation of mouse preantral follicles followed by vitrification in presence of low concentrations of cryoprotectants and laser- and electromagnetic-field assisted controlled ultra-rapid rewarming of capsules with nanoparticles to support and improve fertility preservation and survival of follicles. The paper presents convincing evidence that oocytes grown subsequently to antral stages and in vitro matured to metaphase II are of high quality and developmental potential and can develop to healthy pups after in vitro fertilization and transfer of embryos to foster mothers and that healthy offspring can be obtained from these after mating in vivo.

Overall, the study in the mouse model provides interesting and convincing data on the feasibility to use ultra-rapid nano-warming techniques in vitrification with low cryoprotectant to improve survival and vitality of preantral follicles in fertility preservation that might also open new approaches for restoring fertility in human.

Unfortunately, the presentation of data is not concise and it is difficult for readers to retrieve the main message about experimental set-up and rationale of using the mouse model. Furthermore, background of up-to-date knowledge on human fertility preservation and use of terms defining different stages of folliculogenesis are not precise or misleading and need to be revised, including citing more recent recommendations and up-to-date reports and reviews. Therefore, I cannot recommend accepting for publication in the present form but only after major revision.

For instance, recent advances in cryopreservation have addressed the potential harm to biological samples by ice crystal formation and multiple new approaches to increase efficiency and safety of cryopreservation (e.g. Chang and Zhao, Adv Sci 2021). These include use of different types of ice-inhibiting substances and membrane permeable or impermeable polymers, bioinspired cryoprotecting agents, hydrogels, nanoparticle delivery of cryoprotectants or nanocoating, e.g. with bioinspired materials and physical methods like magnetic, electro-magnetic field or laser thawing. The study has focused on preventing ice-crystal formation and damage during the re-warming procedure in presence of only low concentrations of cryoprotecting agents and is of relevance and interest.

Provided a revision is carefully conducted I highly recommend that the paper should be once more submitted and re-considered for publication in Nature Communications as it presents an innovative, novel concept for potentially successful novel methods that might be of interest in preservation of human and animal fertility. I would be happy to see it again for review as it has been carefully conducted using up-to-date novel methods to improve fertility preservation to scientists and clinicians.

[Response]: We thank the reviewer for these encouraging comments. We have

thoroughly revised and modified the main text to address your concerns, and detailed corrections are listed below point by point. Hopefully, you will find it justified.

Specific Comments:

1. Title should be shorter and more concise, e.g.:Efficient cryopreservation in presence of low cryoprotectants of hydrogel micro-encapsulated mouse preantral follicles by novel rapid nano-warming techniques.

[A1]: Thank you for your suggestion. As per advice, the title was revised in this manuscript (Line 1-3, page 1 in this revision).

2. Abstract:

From the beginning it should be made clear that the major source of oocytes in mammalian ovary is in the primordial follicle pool that becomes depleted during aging or after gonadotoxic exposures. Such follicles are activated in vivo in a complex process and may also become activated inadvertently by stressors such as by freezing/ warming/ mechanochemical stressors. Homogenous populations of preantral follicles that possess a GV oocyte surrounded by one or few layers of granulosa cells can be conveniently obtained from hormonally stimulated prepubertal mice, or by young pubertal females and are therefore often used as a model for optimizing cryopreservation, in vitro culture conditions and fertility preservation in fertility research (Heiligentag et al., *Reprod Fertil Dev.* 2017; Xu and Zelinski, *Biol Reprod* 2021). Preantral follicles are also common in ovarian cortex of adult large mammals and used for preservation but not routinely used in human treatment.

[A2]: Thank you for your valuable suggestion. As per advice, the related information: PAFs could be easily obtained from female animals and were often used as animal models in fertility preservation was added in the abstract in this manuscript as follows (Lines 28-29, page 2 in this revision).

Preantral follicles (PAFs) are often used as a model for cryopreservation and in vitro culture due to their easy availability.

3. Line 29: It is not mainly the cryopreservation of preantral but rather primordial follicles- preferably in ovarian tissue cryopreservation that is of relevance for fertility preservation in humans (e.g. Szymanska et al., *Mol Hum Reprod* 2020, Dolmans et al., *J Clin Med.* 2021; ESHRE Guideline: female fertility preservation Anderson et al., *U Hum Reprod Update* 2020). Better state: for cryopreservation..is one promising..of fertility in mammals.

[A3]: Thank you for your valuable suggestion. As per advice, the sentence was revised in this manuscript as follows (Lines 29-32, page 2 in this revision).

As a promising approach for mammalian fertility preservation, vitrification of PAFs requires extremely high concentrations of penetrating cryoprotective agents (pCPAs; highly toxic organic solvents; up to 6 M).

4. Success using follicle culture is difficult in human due to the long period of folliculogenesis and the dramatic increase in size of maturing follicles not only in human but also other larger species like human, bovine, porcine or macaque. As stated by Gandoli et al., (Animal Reprod. 2019) rather “the use of ovarian fragments is the method for fertility preservation in human”. Also, multiple step procedures are currently used in more experimental approaches which mainly involve primordial follicle activation and growth in slices, isolation of cumulus-oocyte complexes from large antral stages and in vitro maturation rather than preservation of pre-antral follicles (Telfer and Andersen, Fertil Steril 2021). Therefore, authors should from the beginning state that their aims of the present study is to improve cryopreservation techniques and avoid exposures to high cryoprotectants and mechanical and oxidative stress. Apparently, their interest is mainly in using the mouse model of preantral follicle culture to improve methods of cryopreservation by vitrification and advance encapsulation supporting good survival with a novel strategy to support follicle growth and oocyte maturation, a method that might have a potential to be applied also in human fertility preservation in the future.

[A4]: Thank you for your suggestion. The preservation of ovarian fragments is a viable method for human fertility preservation. Here, we mainly used the PAFs of mice as a model, and achieved low-pCPA vitrification through hydrogel encapsulation and nano-warming to improve the cryopreservation effect of PAFs. A new strategy was found to better support PAF growth and oocyte maturation which might also have potential applications in preserving human fertility in the future. As per advice, the related information was revised in this manuscript as follows (Lines 28-34, page 2 in this revision).

Preantral follicles (PAFs) are often used as a model for cryopreservation and in vitro culture due to their easy availability. As a promising approach for mammalian fertility preservation, vitrification of PAFs requires extremely high concentrations of penetrating cryoprotective agents (pCPAs; highly toxic organic solvents; up to 6 M). Here, we successfully accomplished low-pCPA (1.5 M) vitrification of mouse PAFs encapsulated in hydrogel by nano-warming.

5. Line 34: Better to state order of protocol: microencapsulation and vitrification followed by nonwarming using ...to achieve extremely rapid re-warming to prevent ice-crystal formation.

[A5]: Thank you for your suggestion. As per advice, the sentence was revised in this manuscript as follows (Lines 32-34, page 2 in this revision).

Here, we successfully accomplished low-pCPA (1.5 M) vitrification of mouse PAFs encapsulated in hydrogel by nano-warming.

6. Line 35: It is also common for conventional vitrification of murine or larger species PAFs to achieve close to or over 90% survival (e.g. Trapphoff et al., Hum Reprod. 2010) state at which time after thawing the assays were performed and what percentage of

recovered follicles produces MII oocytes after culture. In ovarian tissue cryopreservation follicles survived up to 98% according to other recent reports (El Cury-Silva et al., Cryobiology 2021). It is more important to state how many of the follicles yielded mature oocytes, pups and offspring after IVF, and compare this to controls using conventional vitrification methods.

[A6]: Thank you for your suggestion. As per advice, the related information was added in the abstract in this manuscript as follows (Lines 34-41, page 2 in this revision). And the comparison with controls using conventional vitrification methods was discussed in the discussion.

Compared with conventional water bath warming, the survival rate of PAFs was increased by 33% (90% vs. 57%). The cavity formation rate of hydrogel micro-encapsulated PAFs after in vitro culture was comparable to the control group (no vitrification) (13% vs. 17%). And the percentage of MII oocytes in the cumulus oocyte complexes (COCs) collected from the antral follicles and the birth rate of offspring after in vitro fertilization (IVF) and embryo transfer were also similar to the control group (66.5% vs. 68.0% and 59.9% vs. 60.9%).

7. Line 36: How many of these developed from the surviving follicles?

[A7]: Thank for your suggestion. As per advice, the related information was added in the abstract in this manuscript as follows (Lines 34-38, page 2 in this revision).

Compared with conventional water bath warming, the survival rate of PAFs was increased by 33% (90% vs. 57%). The cavity formation rate of hydrogel micro-encapsulated PAFs after in vitro culture was comparable to the control group (no vitrification) (13% vs. 17%).

Introduction:

8. Line 51: Use more recent citations on increase in cancer risks such as: reports from the UK: Cancer Research UK Breast Cancer Incidence Statistics. [(accessed on 24 April 2019)]; Available online: <https://www.cancerresearchuk.org/health-professional/cancer-statistics/s>.

[A8]: Thank you for your suggestion. The citation was added in this manuscript (Line 47, page 3 in this revision).

9. Line 54: Again, cite more recent literature (e.g. Szymanska et al., Mol Hum Reprod, 2020; Oktay et al., Fertil Steril 2022; Cacciottola et al., Best Pract Res Clin Obstet Gynaecol. 2021)

[A9]: Thank you for your suggestion. The citations were added in this manuscript (Line 50, page 3 in this revision).

10. Line 58: Missing relevant recent citations! (ESHRE Guidelines fertility preservation, Anderson et al., Hum Reprod Open. 2020, Dolmans et al., J Clin Med. 2021 Etc.)

[A10]: Thank for your suggestion. The citations were added in this manuscript (Line 54, page 3 in this revision).

11. Line 71: This is simply not true! The cortex of ovary is particularly rich in primordial follicles and might be preserved in slices which can be transferred or in vitro cultured to obtain larger follicles from which mature IVM oocytes can be obtained (see for instance: discussion by Vo and Kuvamura, J Clin Med 2021).

[A11]: Apologize for the negligence and thank for your suggestion. Primordial follicles composed of an oocyte surrounded by one layers of granulosa cells. Preantral follicles composed of an oocyte surrounded by more than two layers of granulosa cells. Preservation of ovarian tissue slices containing primordial follicles is feasible for cryopreservation of human fertility. Here we mainly used the PAFs of mice as a model to improve the preservation effect of low CPA vitrification. Besides, the feasibility of mouse PAFs cryopreservation has been reported in the literature. For example, Bus A. et al. have mentioned “Early PAFs account for the vast majority of follicles in the ovarian cortex. The majority of these (99.9%) will never mature into preovulatory follicles (Morita and Tilly, 1999), but rather will perish at a premature stage along the developmental path. The stock of primordial and primary follicles thus represents an untapped potential, which could be cultivated for reproduction, preservation, or research purposes.” in the article “Is the pre-antral ovarian follicle the ‘holy grail’ for female fertility preservation?”. To avoid confusion, the sentence in the manuscript was revised as follows (Lines 66-73, page 6 in this revision).

For these women, cryopreservation of isolated ovarian follicles and in vitro culture is an alternative and safe option¹⁷. Preantral follicles (PAFs) are the early stage of follicles, which are composed of an oocyte surrounded by one or few layers of granulosa cells¹⁸. PAFs were abundant and easily available in hormone-stimulated prepubertal mice or young adolescent female mice¹⁹. Therefore, mouse PAFs were often used as models for optimizing cryopreservation, in vitro culture conditions, and fertility preservation in fertility studies^{18,19}.

17. He, X. Microfluidic encapsulation of ovarian follicles for 3D culture. *Ann Biomed Eng* 45, 1676–1684 (2017).

18. Heiligentag, M., Eichenlaub-Ritter, U., Heiligentag, M. & Eichenlaub-Ritter, U. Preantral follicle culture and oocyte quality. *Reprod. Fertil. Dev.* 30, 18–43 (2018).

19. Gupta, P. S. P. et al. Effect of different vitrification protocols on post thaw viability and gene expression of ovine preantral follicles. *Theriogenology* 178, 1–7 (2022).

12. Line 72: It is not preantral but primordial follicles that are the largest population in the ovary. Primordial follicles need to become activated to develop into pre-antral and antral follicles. Most of preantral follicles that become activated in vivo undergo demise and can therefore be rescued for fertility preservation.

[A12]: Thanks for the distinction between PAF and primordial follicles. As per advice, the sentence in the manuscript was deleted.

13. Line 74: During primordial follicle ..

The authors should provide much better insights into procedures that can be used in fertility preservation, in particular in larger mammals. While it is possible to obtain large numbers of pre-antral follicles in prepubertal animals like the mouse after hormonal stimulation, this is certainly not possible in human. Such follicles can be generated from ovarian slices and subsequently cultured to larger stages from which oocyte cumulus-complexes can be derived and oocytes matured *in vitro* to metaphase II. However, the cryopreservation of preantral follicles would only be performed after activation of primordial follicles in slices of ovarian tissue, or is used to rescue *in vivo* activated PAFs that may undergo atresia and is not common practice in human fertility preservation protocols.

[A13]: Thank you for your suggestion. Indeed, our paper focused on studying mouse PAFs as a model for cryopreservation and *in vitro* development. The cryopreservation of PAFs for large mammals and even humans is an important content that needs further study in the future.

14. Line 81 to 84: Again, authors do not correctly use the term preantral and primordial follicles. The latter are the most abundant whereas the formers are limited in numbers and present a larger sub-population of follicles that are not arrested but activated, can potentially mature but usually mostly become atretic. For animal fertility preservation activation to form preantral follicles can be used to isolate preantral stages however in pre-pubertal girls follicles may be isolated after activation in culture of slices.

[A14]: Thanks for the distinction between PAF and primordial follicles. As per advice, the sentences were deleted in the manuscript (Lines 81-84, page 6 in this revision).

15. Line 122: Add reference: Chang and Zhao, Adv Sci (Weinh) 2021

[A15]: Thank for your suggestion. As per advice, the reference was added in the manuscript (Line 109, page 5 in this revision).

16. Line 129: oocytes and granulosa cells

[A16]: Apologize for the negligence, and thank you for your suggestion. As per advice, the words were revised in the manuscript (Line 116, page 6 in this revision).

17. Line 139: preovulatory antral follicles

[A17]: Apologize for the negligence, and thank you for your suggestion. As per advice, the words were revised in the manuscript (Lines 124-125, page 8 in this revision).

Results:

18. It is difficult for readers to follow the procedures that are used from start to completion. Thus, Fig 1 and text does not contain information on encapsulation prior to vitrification and vitrification procedures in experimental pAFs and controls. It appears that an open straw method has been used but it is difficult to retrieve the information whether EG, DMSO or other agents have been used for follicle vitrification

in the control and sample. One has to search methods and read subsequent paragraphs to find the details of the experimental procedures before and during vitrification. Although the paper is focused on the re-warming steps, the information on how vitrified PAFs are obtained needs to be given first. It is only evident from Fig. 7 that PAF-alginate encapsulation has been performed before follicles were vitrified. Therefore, the paper would greatly benefit from a flow chart that might be added as supplement or better at beginning of the result section providing information on culture medium, alginate concentration and cryoprotectants before and during vitrification and on all washing solutions and rewarming steps used during the re-warming process.

[A18]: Thank you for your suggestion. As per advice, a flow chart of encapsulation, cryopreservation, and subsequent development of preantral follicles (PAFs) were added in **Figure 1**, and the explanation was added in this manuscript (Lines 151-183, page 8-10 in this revision).

Encapsulation, Cryopreservation, in vitro Development of PAFs, IVF and transplant

A schematic diagram of the encapsulation, cryopreservation and subsequent development of PAF is shown in Fig. 1. PAF-encapsulated hydrogel microcapsules were generated by dropping a 1% (w/v) alginate solution containing PAFs into a 0.15 M CaCl₂ solution through a centrifugal microfluidic device (Fig. 1a). The particle size of the hydrogel microcapsules varied slightly at different centrifugation rates. Moreover, the morphology of the lyophilized hydrogel microspheres did not change before and after freezing (Supplementary Fig. 1). The PAF-encapsulated hydrogel microcapsules were transferred into CPA (0.75 M E₃G + 0.75 M PROH + 1 M trehalose) supplemented with NPs (0.3% Fe₃O₄ and 0.03% GO) and incubated at 4°C for 10–15 min. Aspirate them into a 2.5 mL straw and seal the straw with wax oil. The straw was then quickly immersed in liquid nitrogen (LN₂) for vitrification. After 24 hours, the straw was removed from LN₂ and quickly immersed in a 50 mL centrifuge tube filled with 37°C water and placed in an electromagnetic coil, while the laser transmitter was turned on for rapid warming (Fig. 1b). After thawing, the solution in the straw was transferred to a 6 mm dish containing washing solution (MEM medium supplemented with 10% FBS). The NPs were drawn to the edge of the petri dish by a magnet and then removed using a pipette. Fe₃O₄ nanoparticles (NPs) in the solution were drawn to the edge of the dish by a magnet and then removed using a pipette (Fig. 1b). The hydrogel microcapsules were siphoned and transferred to a 96-well plate for another wash. The hydrogel microcapsules were then transferred to a 96-well plate via a siphon and subjected to one more wash. The PAFs were cultured in 96-well plates and developed into antral follicles on day 13. Next, the hydrogel microcapsules were dissolved by 75 mM sodium citrate and antral follicles were collected. Antral follicles were punctured via a 1 mL syringe to release the cumulus oocyte complexes (COCs). Oocytes and granulosa cells were separated by hyaluronidase (Fig. 1c). MII stage oocytes were selected for IVF. 2-cell stage embryos were transplanted into pseudopregnant recipient and produced healthy offspring (Fig. 1d).

19. Line 151: Figure 1: Explain abbreviation in figure legend in this as in all other figures: e.g. NPs: nanoparticles, GO etc

[A19]: Apologize for the negligence and thank you for your suggestion. As per advice, the explained abbreviation was added in **Fig. 1** legend in this manuscript (Lines 147-

149, page 8 in this revision).

20. Line 161: This information should be given in abstract to make clear that the paper compared the survival and developmental capacity of alginate encapsulated PAFs that were re-warmed by different methods to prevent ice-crystal formation under low CPA concentrations.

[A20]: Thank for your suggestion. As per advice, the specific data was added in the Abstract in this manuscript as follows (Lines 34-41, page 2 in this revision).

Compared with conventional water bath warming, the survival rate of PAFs was increased by 33% (90% vs. 57%). The cavity formation rate of hydrogel micro-encapsulated PAFs after in vitro culture was comparable to the control group (no vitrification) (13% vs. 17%). And the percentage of MII oocytes in the cumulus oocyte complexes (COCs) collected from the antral follicles and the birth rate of offspring after in vitro fertilization (IVF) and embryo transfer were also similar to the control group (66.5% vs. 68.0% and 59.9% vs. 60.9%)..

21. Line 166: may? Better: to explore improved vitality and developmental competence of cryopreservation with low cryoprotectant.

[A21]: Apologize for the negligence, and thank you for your suggestion. **Fig. 1** and the corresponding explanation were rewritten, and this sentence was removed.

22. Line 252: (0,green, 1M light green, 1.5M yellow, 2M blue bars)..different CPAs (vertical axis)..

[A22]: Apologize for the negligence and thank you for your suggestion. As per advice, the sentence was revised in this manuscript as follows (Lines 263-271, pages 14-15 in this revision).

(b) Viability of the PAFs after vitrification with different CPAs (0 M, green; 1 M, light green; 1.5 M, orange; 2 M, purple bar). (c) Viability of the PAFs encapsulated in different concentrations of alginate hydrogel (0%, green; 1%, rust; 1.5%, pink; and 2%, emerald green bars. w/v) after vitrification with different 1.5 M CPA. (d) Viability of PAFs after warming under different conditions (Fresh: green; W/O Encap, MIH and LIH: light green; W/ Encap: orange; W/ Encap and MIH: pink; W/ Encap and LIH: emerald green; W/ Encap, MIH and LIH: blue bars).

23. Line 254: Unclear legend. Please, refer to color in groups.

[A23]: Apologize for the inconvenience and thank you for your suggestion. As per advice, the sentence was revised in this manuscript as follows (Lines 263-271, pages 14-15 in this revision).

(b) Viability of the PAFs after vitrification with different CPAs (0 M, green; 1 M, light green; 1.5 M, orange; 2 M, purple bar). (c) Viability of the PAFs encapsulated in different concentrations of alginate hydrogel (0%, green; 1%, rust; 1.5%, pink; and

2%, emerald green bars. w/v) after vitrification with different 1.5 M CPA. (d) Viability of PAFs after warming under different conditions (Fresh: green; W/O Encap, MIH and LIH: light green; W/ Encap: orange; W/ Encap and MIH: pink; W/ Encap and LIH: emerald green; W/ Encap, MIH and LIH: blue bars).

24. Line 323: Rate of MII for all cultured follicles or only those maturing to antral stage?? Please, add numbers to each panel

[A24]: Thank you for your suggestion. **Fig. 4g** mainly showed the rate of MII for the follicles maturing to the antral stage. As per advice, the sentence was revised in this manuscript (Lines 349-350, page 19, Lines 359-361, page 20 in this revision), and sample numbers were added in **Fig. 4g** legends.

(f) Percentage of MII oocytes in the collected COCs.

n=10-20 for three replicates (b); n=30 for three replicates (c); n=40-50 for four replicates (e); n=10-20 for four replicates (f); n=10-20 for five replicates (h). n: number of follicles used in each experiment.

25. Line 334: .. appeared in % of PAFs.. % of PAFs (n=...respectively)

[A25]: Thank you for your valuable suggestion. The percentage of antral follicles in all cultured follicles was shown in **Fig. 4e**, so specific rates were added in this manuscript as follows (Lines 384-386, page 21 in this revision).

The percentage of antral follicles post-vitrification (13%) was also lower than that of the W/O VTF group (17%) (Fig. 4e).

26. Line 351: cultured pre-antral follicles?? Of similar size??

[A26]: Apologize for the inconvenience and thank you for your suggestion.

i. As per advice, the sentence was revised in this manuscript as follows (Lines 387-389, page 21 in this revision).

To further evaluate the quality of antral follicles, we obtained oocytes at the MII stage after culture of PAFs in vitro.

ii. The sizes of MII oocytes from Fresh, W/VTF and W/MIH+LIH were similar (about 75~80 μm).

27. Line 355: Omit this sentence. In the study it is not the MII oocytes that are vitrified warmed and therefore formation of spindles is just reflecting normal maturation of oocytes after culture of follicles.

[A27]: Thank you for your suggestion. As per advice, the sentences “The spindle is easily disrupted during cooling and warming due to changes in temperature and ice crystal formation. The spindle breakage usually causes chromosomal disorganization, which adversely affects the euploidy of the embryo after fertilization.” were deleted in

this manuscript.

28. Line 370: Better: Functional mitochondria are essential for normal oogenesis and embryogenesis.

[A28]: Thank you for your suggestion. As per advice, the sentence was revised in this manuscript as follows (Lines 404-405, page 22 in this revision).

*Functional mitochondria are essential for normal oogenesis and embryogenesis*⁵⁰.

50. Iwata, H. Resveratrol enhanced mitochondrial recovery from cryopreservation-induced damages in oocytes and embryos. *Reproductive Medicine and Biology* 20, 419–426 (2021).

29. Line 372: add more recent citations such as De Cota et al., *Reprod Fert Dev*2020; Iwata , *Reprod. Med Biol* 2021

[A29]: Thank you for your suggestion. As per advice, the references were added in this manuscript (Lines 405, page 22 in this revision).

30. Line 382: Add information on which oocytes have been analyzed: The GV oocytes after re-warming or rather MII oocytes after culture? This is vital as MMP may transiently decrease and then recover; see and add citation: Demand et al., 2012

[A30]: Apologize for the negligence, and thank you for your suggestion. As per advice, the information was added to this manuscript as follows (Lines 413-414, page 22 in this revision).

Hence, the MMP of MII oocytes was detected by JC-1 staining in this study.

31. Line 393 and following: always state stage of oocytes, GV or MII, from re-warming or after culture and growth?

[A31]: We thank the reviewer for valuable comment.

i. The evaluation of MMP, ATP, MS, ROS and Calcium ions levels were always for MII stage oocytes.

ii. The MII oocytes were come from Fresh (MII stage oocytes obtained directly from mice), W/O VTF (PAFs encapsulated in hydrogel microcapsules were directly cultured in antral follicles *in vitro* without vitrification and warming, and then MII stage oocytes were obtained from antral follicles), W/ MIH+LIH (PAFs encapsulated in hydrogel microcapsules were cultured into antral follicles *in vitro* after vitrification and warming, and then MII stage oocytes were obtained from the antral follicles).

32. Line 415, Fig. 5: Alter blastoshere to blastocyst

[A32]: We thank the reviewer for valuable comment. As per advice, the word was revised in **Fig. 5**.

33. Figure 6c: There appear to be always twice as many male compared to female pups after 2-cell transfer- please explain!

[A33]: We thank the reviewer for valuable comment. Due to the relatively small number of previous experiments, a male-to-female ratio of 2:1 appeared. So, several more transplantation experiments were performed, and the data was organized, and then the results were revised in **Fig. 7c**.

34. Line 423: needs to be..

[A34]: Thank you for your valuable comment. As per advice, the sentence was revised in this manuscript as follows (Lines 463-464, page 25 in this revision).

To examine the developmental potential of MII oocytes, IVF needs to be performed.

Discussion:

35. Line 499: This is a most relevant improvement that should be pointed out in the abstract

[A35]: Thank you for your valuable suggestion. As per advice, the sentence was moved to the abstract in this manuscript (Lines 34-36, page 2 in this revision).

Compared with conventional water bath warming, the survival rate of PAFs was increased by 33% (90% vs. 57%).

36. Line 502: only with low CPA- traditional warming with high CPA and washing steps has much higher survival rates e.g.

[A36]: Thank you for your suggestion. The literature has reported that the survival rate of mouse PAFs could reach 84.1% within 4 days after traditional high CPA vitrification and water bath warming and the survival rate of the control group (non-vitrified group) was 90.8%¹. In this study, the survival rate of PAFs could reach 90% within 6 hours after low-CPA vitrification and nano-warming, which did not change within 4 days. Therefore, our method could well improve the post-freezing survival rate of PAFs. As per advice, the sentences were added in the discussion in this manuscript as follows (Lines 563-566, page 32 in this revision).

In addition, compared with high-CPA vitrification and water bath warming reported in previous literature⁶⁰, the low-CPA vitrification and nano-rewarming can effectively improve the post-freeze survival rate of PAFs

60. Trapphoff, T., El Hajj, N., Zechner, U., Haaf, T. & Eichenlaub-Ritter, U. DNA integrity, growth pattern, spindle formation, chromosomal constitution and imprinting patterns of mouse oocytes from vitrified pre-antral follicles. *Human Reproduction* 25, 3025–3042 (2010).

37. Line 527: This is a comparative short time for toxicity test and remaining NP should be analyzed in future studies.

[A37]: Thank you for your valuable suggestion. As per advice, in the toxicity test of NPs, the incubation time of NPs with PAFs was increased up to 60 h. The result was

added to this manuscript (Lines 239-244, page 13 in this revision).

38. Line 538: It may depend on size of follicles in different species whether microencapsulation is favorable or restricts follicle and oocyte expansion and growth.

[A38]: We thank the reviewer for valuable comment. The mouse PAFs used in this study were small (120-190 μm in diameter), and the low concentration of alginate hydrogel not only maintained the 3D morphology of follicles but also facilitated the expansion and growth of follicles and oocytes.

39. Line 558: this may be true for the mouse model- however, follicle growth in larger mammals may be not so ideal when there is more mechanical stress and possible problems with oxygen diffusion.

[A39]: Thank you for your suggestion. Some studies have pointed out that low alginate concentration is beneficial to the development of mouse PAF, so this study selected 1% alginate-encapsulated mouse PAF for subsequent culture¹. The concentration of hydrogels most suitable for *in vitro* development of PAFs in larger mammals is important content for our future research.

1. Jamalzaei, P., Valojerdi, M. R., Montazeri, L. & Baharvand, H. Effects of alginate concentration and ovarian cells on *in vitro* development of mouse preantral follicles: a factorial study. *Int J Fertil Steril* 13, 330–338 (2020).

40. Line 572: the MII oocytes

[A40]: Apologize for the negligence and thank you for your suggestion. As per advice, the word was revised in this manuscript (Line 618, page 34 in the revision).

41. Line 591: Omit sentence. The successful..Better: Adaptation of the novel protocols for cryopreservation for human ovarian material and PAFs may help improve...

[A41]: Thank you for your suggestion. As per advice, the sentence was revised in this manuscript as follows (Lines 640-642, pages 35-36 in the revision).

Adaptation of the novel protocols for cryopreservation for human ovarian material and PAFs may help improve chances of regaining fertility for patients after treatment.

M&M:

42. Line 616: of ?? week old ..

[A42]: We thank the reviewer for valuable comment. As per advice, the sentence was revised in this manuscript (Lines 663-664, page 38 in this revision).

PAFs (100-190 μm) were isolated from the ovaries of 3-week-old KM mice by mechanical separation.

43. Line 706: Which concentration of CPA was used?

[A43]: We thank the reviewer for valuable comment. As per advice, the sentence was revised in this manuscript (Lines 757-758, page 42 in this revision).

In this work, the CPA solution consisted of pCPAs (0.75 M EG and 0.75 M PROH) and nonpenetrating CPAs (1 M trehalose).

44. Line 726: removed by??

[A44]: Thank you for your suggestion. As per advice, the sentence was revised in this manuscript (Lines 776-779, page 43 in this revision).

After washing three times by α -MEM medium, the CPA solution was removed. And then the microcapsules were kept in separation medium at 37°C for subsequent experiments.

45. Line 898: 6 embryos each??

[A44]: Thank you so much for your suggestion. As per advice, the sentence was revised in this manuscript (Lines 891-892, page 48 in this revision).

An incision was made in the fallopian tube, and embryos (9-16) were injected into the fallopian tube through a microtubule at a time.

46. Consider to add references, e.g.

Cancer Research UK Breast Cancer Incidence Statistics. [(accessed on 24 April 2019)]; Available online: <https://www.cancerresearchuk.org/health-professional/cancer-statistics/s>.

Cacciottola L, Donnez J, Dolmans MM. Best Pract Res Clin Obstet Gynaecol. 2021 Nov 15;S1521-6934(21)00165-6. doi: 10.1016/j.bpobgyn.2021.09.010. Online ahead of print. PMID: 34887172

El Cury-Silva T, Nunes MEG, Casalechi M, Comim FV, Rodrigues JK, Reis FM. Cryobiology. 2021 Dec;103:7-14. doi: 10.1016/j.cryobiol.2021.08.001. Epub 2021 Aug 8. PMID: 34370991

ESHRE Guideline Group on Female Fertility Preservation, Anderson RA, Amant F, Braat D, D'Angelo A, Chuva de Sousa Lopes SM, Demeestere I, Dwek S, Frith L, Lambertini M, Maslin C, Moura-Ramos M, Nogueira D, Rodriguez-Wallberg K, Vermeulen N. Hum Reprod Open. 2020 Nov 14;2020(4):hoaa052. doi: 10.1093/hropen/hoaa052. eCollection 2020. PMID: 33225079

Del Castillo LM, Buigues A, Rossi V, Soriano MJ, Martinez J, De Felici M, Lamsira HK, Di Rella F, Klinger FG, Pellicer A, Herraiz S. Hum Reprod. 2021 Aug 18;36(9):2514-2528. doi: 10.1093/humrep/deab165. PMID: 34333622

Telfer EE, Andersen CY. Fertil Steril. 2021 May;115(5):1116-1125. doi: 10.1016/j.fertnstert.2021.03.004. Epub 2021 Apr 3. PMID: 33823993

Rodriguez-Wallberg KA, Marklund A, Lundberg F, Wikander I, Milenkovic M, Anastacio A, et al. . Acta Obstet Gynecol Scand (2019) 98(5):604 – 15. doi: 10.1111/aogs.13559 - DOI - PubMed

Laronda MM, Rutz AL, Xiao S, Whelan KA, Duncan FE, Roth EW, Woodruff TK, Shah RN. Nat Commun. 2017;8:15261. doi: 10.1038/ncomms15261.

Ting AY, Yeoman RR, Campos JR, Lawson MS, Mullen SF, Fahy GM, Zelinski MB. Hum Reprod. 2013;28(5):1267 – 1279. doi: 10.1093/humrep/det032.

Moravek MB, Appiah LC, Anazodo A, Burns KC, Gomez-Lobo V, Hoefgen HR, Jaworek Frias O, Laronda MM, Levine J, Meacham LR, Pavone ME, Quinn GP, Rowell EE, Strine AC, Woodruff TK, Nahata L.. J Adolesc Health. 2019;64(5):563 – 573. doi: 10.1016/j.jadohealth.2018.10.297.

Szymanska et al., Mol Hum Reprod 2020 Aug 1;26(8):553-566.
doi: 10.1093/molehr/gaaa043.

Kim et al. Theriogenology 2020 Mar 1;144:33-40.
doi: 10.1016/j.theriogenology.2019.12.009.

Sugishita et al Hum Reprod 2021 Dec 20;deab274.
doi: 10.1093/humrep/deab274. Online ahead of print.

Gupta et al. Theriogenology . 2022 Jan 15;178:1-7.
doi: 10.1016/j.theriogenology.2021.10.024. Epub 2021 Oct 25.

Wu et al. Int J Biol Macromol 2021 Dec 1;192:1276-1291.
doi: 10.1016/j.ijbiomac.2021.09.211. Epub 2021 Oct 9.

Chang and Zhao Adv Sci (Weinh) 2021 Feb 1;8(6):2002425.
doi: 10.1002/advs.202002425. eCollection 2021 Mar.

Delattre et al., . 2020 Nov 1;35(11):2524-2536.
doi: 10.1093/humrep/deaa193.

We would like to thank the reviewer again for taking the time to review our manuscript.

Reviewers' Comments:

Reviewer #1:

Remarks to the Author:

The manuscript was revised accordingly and is much improved. All my concerns have been appropriately addressed. In my opinion, the manuscript can now be accepted in its present form.

Reviewer #3:

Remarks to the Author:

The revision has addressed all concerns of both reviewers and significantly improved the paper. Since it provides novel, concise and significant information from the mouse preantral follicle culture as a model that can be used to improve cryopreservation/vitrification of mammalian follicles, I highly recommend accepting this excellent paper for publication.

The paper now contains all additional information requested, e.g. on specifying M&M, statistical analysis and changes in the text to avoid any ambiguities and improve precision.

There are only two minor changes that should be made:

Line 374: change "granular cells" to "granulosa cells"

Line 860-68: Revised sentence could be improved by reading:

To test the mitochondrial., oocytes were divided into five major experimental groups to assess the levels of MMP, ATP, MS, ROS or calcium, respectively (each assay contained three groups...). For this, oocytes were incubated in JC-1 staining....or CellTiter Glo..., or MitoSOX..., or DCDH-DA (...) or, ..staining solution or Fluo-4 AM..., respectively, for 20-30min..Then oocytes..

Response to Reviewers' comments:

We are very grateful to the referees for their insightful and thoughtful comments. In this revision, we have carefully addressed all the comments, which significantly improves the clarity and quality of the manuscript. The point-by-point responses to all the comments are listed below. All changes for addressing the comments are also highlighted in yellow in the revised manuscript

Reviewers' comments:

Reviewer #1:

Remarks to the Author

The manuscript was revised accordingly and is much improved. All my concerns have been appropriately addressed. In my opinion, the manuscript can now be accepted in its present form.

[Response]: We are grateful for the insightful and thoughtful comments. Thank you for agreeing to accept the manuscript.

Reviewer #3:

Remarks to the Author

The revision has addressed all concerns of both reviewers and significantly improved the paper.

Since it provides novel, concise and significant information from the mouse preantral follicle culture as a model that can be used to improve cryopreservation/vitrification of mammalian follicles, I highly recommend accepting this excellent paper for publication.

[Response]: We thank the reviewers for their previous insightful comments. After we revised the text one by one according to your suggestion, the article has been greatly improved. We thank you for agreeing to receive this article.

The paper now contains all additional information requested, e.g. on specifying M&M, statistical analysis and changes in the text to avoid any ambiguities and improve precision.

[Response]: We agree and thank the reviewer's insightful suggestion. We have revised and modified the main text to address your concerns, and detailed corrections are listed below, point by point. We hope that it is now clearer.

There are only two minor changes that should be made:

1. Line 374: change "granular cells" to "granulosa cells"

[A1]: Apologize for the negligence, and thank you for your suggestion. As per the advice, the "granular cells" has been changed to "granulosa cells" in the manuscript (Lines306, page 14 in this revision).

2. Line 860-68: Revised sentence could be improved by reading:

To test the mitochondrial..., oocytes were divided into five major experimental groups to assess the levels of MMP, ATP, MS, ROS or calcium, respectively (each assay contained three groups...). For this, oocytes were incubated in JC-1 staining....or CellTiter Glo..., or MitoSOX..., or DCFH-DA (...) or, ..staining solution or Fluo-4 AM....., respectively, for 20-30min..Then oocytes..

[A2]: We thank the reviewer's insightful suggestion. As per advice, the sentence has been revised in the manuscript (Lines 810-819, pages 38 in this revision)

To test the mitochondrial function of oocytes, oocytes were divided into five major experimental groups to assess the levels of MMP, ATP, MS, ROS or calcium, respectively (each group contained three groups: Fresh, W/O VIT and W/ MIH + LIH). For this, oocytes were incubated in JC-1 staining solution (KGA601, KeyGen, China), CellTiter-Glo Reagent (Luminescent cell viability assay kit, Promega), MitoSOX Red Mitochondrial Superoxide Indicator (M36008, Invitrogen), DCFH-DA (D6470, Solarbio, China) staining solution or Fluo-4 AM (KGAF024, Keygen, China), respectively, for 20-30 min respectively at 37°C in a 5% CO₂ incubator.

We would like to thank the reviewer again for taking the time to review our manuscript.